# Volcano tsunamis and their effects on moored vessels safety: The 2022 Tonga event

Sergio Padilla[1], Íñigo Aniel-Quiroga[1], Rachid Omira[2,4], Mauricio Gonzalez[1], Jihwan Kim[2], Maria A. Baptista[3,4]

[1] Environmental Hydraulics Institute, Universidad de Cantabria (IHCANTABRIA), Avda. Isabel Torres, 15, Santander, Spain

[2] Instituto Português do Mar e da Atmosfera (IPMA), Lisbon, Portugal

[3] Instituto Superior de Engenharia de Lisboa; Lisbon, Portugal

[4] Instituto Dom Luiz, Faculdade de Ciências da Universidade de Lisboa

*Correspondence to*: Sergio Padilla (sergio.padilla@unican.es)

**Abstract.** The explosion of the Hunga Tonga-Hunga Ha'apai volcano on January 15, 2022 (Tonga22) was the origin of a volcano-meteorological tsunami (VMT) recorded worldwide. At a distance exceeding 10,000 kilometres from the volcano and 15 hours after its eruption, the moorings of a ship in the Port of La Pampilla, Callao (Peru), failed, releasing over 11,000 barrels of crude oil.

This study delves into the profound implications of the Tonga 22 event, investigating whether it could have led to the break of the mooring system. We conducted a comprehensive analysis of this significant event, examining the frequency content of the time series recorded at tide gauges, DART buoys, and barometers in the Southern Pacific Ocean. Our findings revealed that the maximum energy of the spectra corresponds to 120 minutes of wave period off the coast of Peru, with the arrival time of these waves coinciding with the time of the accident in the Port.

We used a Boussinesq model to simulate the propagation of the Volcano-Meteorological tsunami from the source to the Port in Peru to study the impact of those waves on the mooring system. We used the synthetic tsunami recorded in the Port as input for the model that simulates mooring line loads based on the ship's degrees of freedom. The results suggest that the 120-minute wave triggered by the VMT could significantly increase mooring stresses due to the resulting hydrodynamic effects, exceeding the Minimum Break Load (MBL).

We conclude that the propagation of the long wave period generated by the VMT caused overstresses in moored lines that triggered accidents in port environments. This event showed the need to prepare Tsunami Early Warning Systems and port authorities for detecting and managing VMTs induced by atmospheric acoustic waves. The work provides new insights into the far-reaching impacts of the Tonga 2022 tsunami.

## 1. Introduction

Volcanic activity can trigger tsunamis through underwater explosion, caldera collapse, pyroclastic flow, flank collapse, or atmospheric gravity waves produced by large explosions (Paris, 2015). Throughout history, tsunamis of volcanic origin have been poorly studied due to their scarcity, leading to a limited understanding of their generation mechanisms and impacts at local, regional, and global scales (Hayward et al., 2022). To better understand this type of tsunami geneses and, therefore, reduce the epistemic uncertainties associated with it, several studies have been carried out (Antonopoulos, 1992; Pararas-Carayannis, 1992, 2004), which also include aquatic environments other than oceans, such as great lakes in Russia and the Philippines (Belousov et al., 2000; Falvard et al., 2018). In these studies, the 1883 eruption of the Krakatau volcano in the Sunda Strait (Indonesia), is commonly accepted as a main reference and a unique example (Kienle et al., 1987), as it was the first event of its kind recorded by different instruments worldwide (Paris et al., 2014; Yokoyama, 1981).

The Hunga Tonga-Hunga Ha'apai volcano (HTHH) is a submarine volcano near an island with the same name, and located 65 km NNW of the island of Tongatapu capital of Tonga, Nukuʻalofa (20.55°S, 175.39°W, see location in Figure 1). It is one of several active volcanoes in the Kingdom of Tonga, an archipelago nation in the South Pacific. The latest eruptive phase of the HTHH volcano began in mid-December 2021 with vigorous shallow-water explosive activity (Vergoz et al., 2022). On January 15, 2022, the volcano erupted at 4:00 a.m. UTC (Tonga Volcanic Eruption & Tsunami, 2022). The HTHH trigger a volcano-meteorological tsunami (VMT) due to a violent volcanic explosion that generated atmospheric gravity waves that propagated several times across the globe (Omira et al., 2022; Wright et al., 2022). These waves resulted from particle agitation in the atmosphere, travelling both vertically and horizontally at sonic and supersonic speeds (Kubota et al., 2022; Matoza et al., 2022; Wright et al., 2022; Dogan et al., 2023). Following the volcano explosion, there were reports of flooding more than 1 m, causing damage to ports and infrastructure in both near and far field (Ramírez-Herrera et al., 2022; Imamura et al., 2022). The affected locations include Australia, New Zealand, the United States, Mexico, and Peru, resulting in economic losses of approximately $102 million due to damages to floating docks, vessels, and infrastructure (Terry et al., 2022; World Bank, 2024).

The Tonga VMT was exceptional as it travelled at faster speeds than common tsunamis, had a global reach, affected the far-field coasts, and caused noticeable damages, human fatalities, and coastal hydrodynamic effects (Terry et al., 2022; Omira et al., 2022; Lynett et al., 2022). For instance in Kochi prefecture 5 boats were sunk and capsized in the port of Sakihama in the city of Muroto (https://oceancrew.org/news/more-than-30-boats-sank-due-to-the-tsunami-in-japan_17-01-2022/), two people died in Peru (INDECI, 2024) and strong currents in ports of Mexico were reported as a direct effects of the tsunami (Ramírez-Herrera et al., 2022). In their study, Omira et al. (2022) demonstrated that the primary source of the globally observed Tonga tsunami was the acoustic gravity waves radiated from the volcanic explosion. Here, the sizeable tsunami at some distant coasts (i.e., South America and Japan) was associated with the amplified ocean waves under Proudman resonance (Proudman, 1929) when the atmospheric wave propagated over very deep water (i.e., oceanic trenches). Other triggering mechanisms, including the submarine volcanic

explosion, likely contributed to the generation of the locally observed tsunami in the far-field (Lynett et al.,
2022; Omira et al., 2022).
The consequences of harbour-intruding long waves on moored vessels have been investigated by numerous
studies (Ayca and Lynett, 2016, 2018, 2021; Kirby et al., 2022; Wilson et al., 2017). Seismic tsunamis often
cause damage to port environments, such as broken moorings, collisions, or subsidence due to the large
amount of momentum flux travelling within the harbour (Lynett et al., 2022; Ohgaki et al., 2008; Inoue et
al., 2001). This also applies to tsunamis induced by atmospheric disturbances, similarly, may have caused
damage to ships, moored vessels, and small bays (Imamura et al., 2022; Thomson et al., 2009). It has been
observed in these studies that strong currents, often accompanying long-period waves, increase the
probability of generating large catastrophes in harbours (Shigeki and Masayoshi, 2009; Sakakibara et al.,
2010; Zheng et al., 2022; Lynett et al., 2012). For instance Shigeki and Masayoshi (2009), studied
numerically the mooring loads due to large-scaled tsunamis in GNL-carriers vessels and found out that the
drag forces due to the currents induce large sway and surge motions are strongly important. Likewise, based
on DOFs (Degrees Of Freedom), some authors such as López and Iglesias (2014) agree with the hypothesis
that the motions in the vessel's horizontal plane (named sway, surge and yaw) are strongly correlated with
the total tsunami wave energy, with the currents being quite significant in ship sway response (Inoue et al.,
2001). Furthermore, Ohgaki et al. (2008) and Zheng et al. (2022) mention that tsunamis are usually closer
to the natural period of a mooring system (>80 sec), which makes these waves more prone to cause damage.
Given the non-linearity of the hydromechanical and physical processes that involve stress studies in
mooring systems, in which each ship has its own characteristics (geometric, inertial, among others), it is
pertinent to perform specific studies focused on each situation, configuration, and need (Zheng et al., 2022).
In La Pampilla Port in Peru, 10,000 km away from the Tonga volcano, the Italian-flagged oil tanker Mare
Doricum reported the breakage of their mooring lines, 15 hours after the explosion of the HTHH volcano.
The ship captain associated the breaking of the vessel's moorings with the abnormal waves in the sea, for
which no warning was issued (SPDA Actualidad Ambiental, 2024). The Peruvian National Tsunami
Warning System (CNAT) stated that the Tonga tsunami did not generate a tsunami on the Peruvian coast
(CNAT, 2022). However, the tide gauge located in Callao Bay recorded a sudden change in sea level
coincident with the time of the accident (UNESCO/IOC, 2021). This article addresses the impact of the
Tonga 2022 tsunami on vessels moored on the Peruvian coast. It uses both sea-level data analysis and
numerical modelling to improve the understanding of the damage caused by far-field Tonga VMT, studying
its influence on the safety of moored vessels.
**2.     Data and Methods**
Considering the complexity of the Tonga tsunami event, which likely involved multiple triggering volcanic
mechanisms, analysing waves measured by both oceanic and atmospheric instruments is highly important
(Wright et al., 2022). Firstly, we used the wavelet analysis to examine the composition of the signals
recorded by both tidal gauges and DART buoys within the Pacific Ocean. Secondly, we studied the
hydrodynamic effects of the Tonga tsunami in the far field, using tsunami numerical simulations
(Boussinesq-type model) over high-resolution bathymetric models. Thirdly, we developed a model to assess
the loads on vessel's mooring lines based on the rigid body analytical equations with six degrees of freedom
(DOFs), this model uses as input the ocean dynamics (ocean elevation and velocities) caused by the VTM.

## 2.1. Air pressure and sea level data

The data used in this study includes records from DART buoys, tide gauges, and weather stations in the
Pacific Ocean. Figure 1 shows the location of the instruments considered in this study, with red circles
representing the atmospheric pressure sensors, light blue triangles for the tide gauges, and yellow dots for
the DART buoys.
The deep-water sea level time series (**D1**, **D2,** and **D3**) were obtained from the DART buoys, which in turn
managed by the Center for Operational Oceanographic Products and Services of the National Oceanic and
Atmospheric Administration (NOAA, https://www.ndbc.noaa.gov/). Coastal sea level data used in this
study come from tidal stations connected in real-time to the Sea Level Station Monitoring Facility of
UNESCO's Intergovernmental Oceanographic Commission (IOC, http://www.ioc-
sealevelmonitoring.org/). The **B1** air pressure data have been obtained from local agents in New Zealand
(NIWA, https://niwa.co.nz/) and those from stations **B3**, **B4**, **B5,** and **B8** came from the Dirección General
De Aeronáutica Civil de Chile (DGAC, https://climatologia.meteochile.gob.cl/) through the Chilean
Meteorological Office.

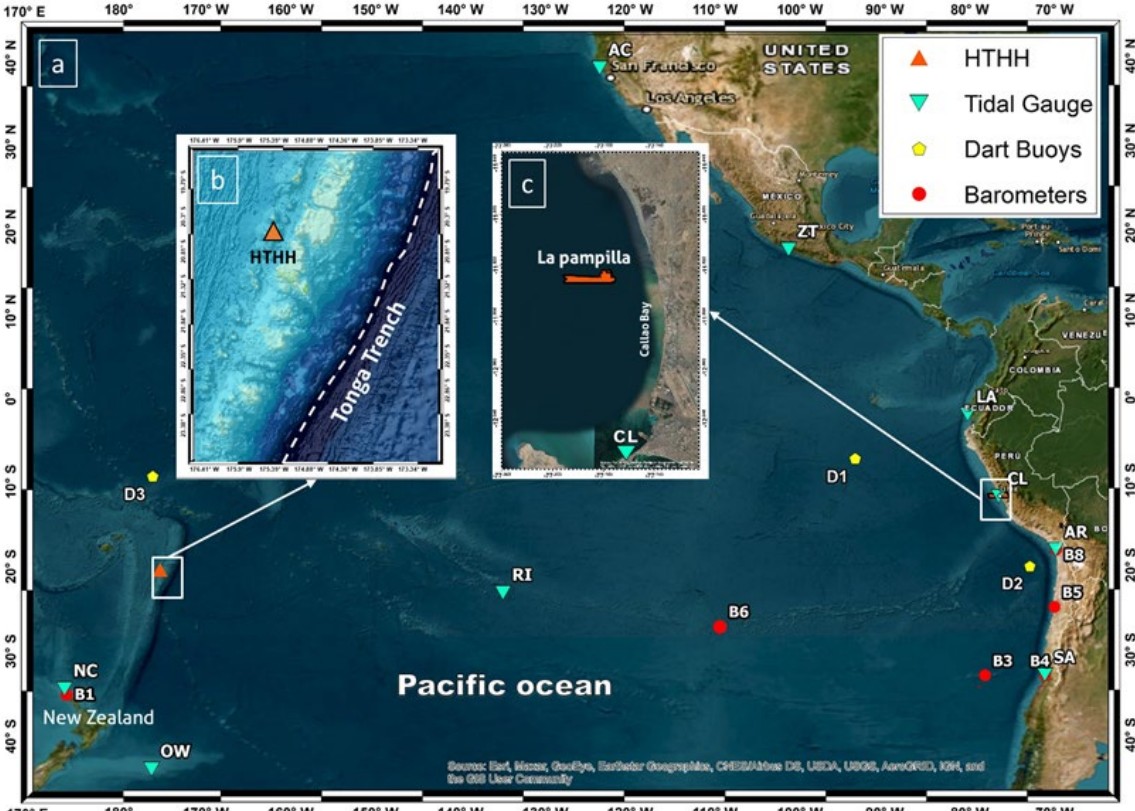


**Figure 1. a, locations of the measuring instruments. Red circles show the atmospheric pressure sensors, the light**
**blue triangles show the tide gauges, and the yellow circles refer to the Dart Buoys. b, shows the HTHH volcano**

 **(orange triangle) and Tonga trench location. c, is the Callao bay zoom, the orange vessel represents the location of terminal 2 of the port of La Pampilla, Peru.**

The sea level time series were de-tided using bandpass filters of 1.5 min-2.5 hours and 2 min - 3 hours for
DART and tide gauge data, respectively, following Lynett et al. (2022). For atmospheric pressure data, a
bandpass filter of 1.5 min - 2.5 hours was used (Table 1). Figure 2 shows the filtered time-series. The arrival
time was less than 3 hours for the **B1** sensor in the near field (shape differences are caused by the sample
time), and between eight and twelve hours in the far field, about 10.000 km away. The air-pressure time-
series shows a notable peak of approximately 2 hPa with the "N-wave" pulse shape associated with the
leading Lamb wave (Lynett et al., 2022; Omira et al., 2022). The seawater elevation, in deep waters Dart
sensors, shows tsunami arrival times in two different moments related with the inverse barometer effect
due to the leading lamb wave and the acoustic gravity waves coupled in ocean (VMT) respectively whit
values no more than 10 cm. On the other hand, the tidal gauge shows tsunami waves values higher than 1
meter especially in the far field locations like Chilean and Peruvian coasts.

| Type | ID | Name | Source (country) | Latitude (deg) | Longitude (deg) | Distance from source (km) | Sample (min) |
|---|---|---|---|---|---|---|---|
| Barometric | B1 | Nort Cape B. | New Zealand Gov | -35.134 | 173.263 | 1750 | 10 |
| | B3 | 330031 | Chilelan Gov. | -33.636 | -78.833 | 9110 | 1 |
| | B4 | 330030 | Chilelan Gov. | -33.656 | -71.613 | 9720 | 1 |
| | B5 | 250005 | Chilelan Gov. | -25.411 | -70.484 | 10230 | 1 |
| | B6 | 270001 | Chilelan Gov | -35.134 | 173.263 | 6530 | 1 |
| | B8 | 180042 | Chilelan Gov. | -18.513 | -70.266 | 10600 | 1 |
| Tide Gauge | NC | North Cape T.G. | IOC (New Zealand) | -34.410 | 173.050 | 1715 | 1 |
| | RI | Rikitea | IOC (France) | -23.118 | -134.969 | 4025 | 1 |
| | AC | Arena Cove | IOC (USA) | 38.913 | -123.705 | 8700 | 1 |
| | ZT | Zihuatanejo | IOC (Mexico) | 17.637 | -101.558 | 9155 | 1 |
| | GL | Galapagos | IOC (Ecuador) | -0.752 | -90.307 | 9410 | 1 |
| | SA | San Antonio | IOC (Chile) | -33.582 | -71.618 | 9720 | 1 |
| | OW | Owenga | IOC (New Zealand) | -44.025 | -176.369 | 2290 | 1 |
| | AR | Arica | IOC (Chile) | -18.476 | -70.323 | 10595 | 1 |
| | CL | Callao | IOC (Perú) | -12.069 | -77.167 | 10240 | 1 |
| | LA | La Libertad | IOC (Ecuador) | 13.485 | -89.319 | 10305 | 1 |
| DART | D1 | 32413 | NDBC (USA) | -7.421 | -93.484 | 8800 | 15 |
| | D2 | 32401 | NDBC (USA) | -20.474 | -73.421 | 10205 | 15 |
| | D3 | 51425 | NDBC (USA) | -9.511 | -176.258 | 1530 | 15 |

**Table 1. Description of measuring instruments.**

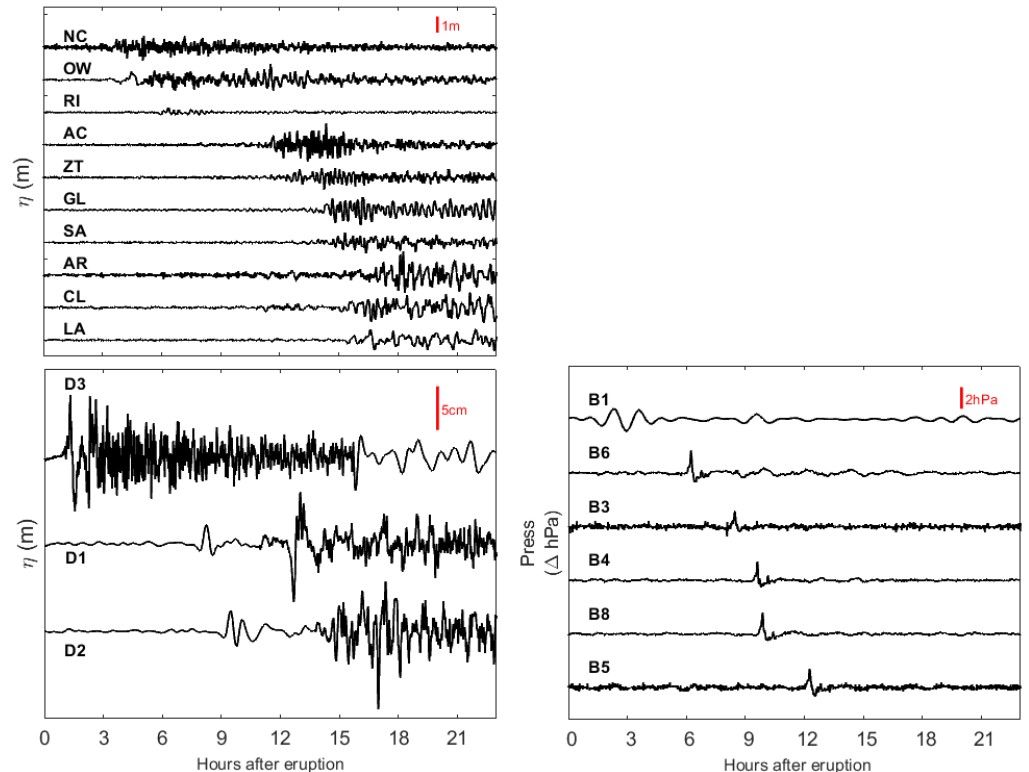


**Figure 2. De-tided time series. Air pressure data (right panel), tide gauge, and Dart data (left panel). Vertical**
**red lines used to scale the magnitude of the variations in each instrument.**
## 2.2. Spectral analysis
Spectral analysis is a practical tool for identifying the characteristic frequencies and energy levels in a time-
series, particularly when it is composed of the superposition of signals with different frequencies. This
method has found extensive application in the study of tsunamis (Abe, 2011; Rabinovich, 1997;
Shevchenko et al., 2011; Satake et al., 2013; Baptista et al., 2016; Xu et al., 2022). In the time domain, we
use wavelets to perform spectral analysis, allowing the estimation of the evolution of the spectral energy
and, consequently, the first instants of energy increase for a given specific period. Wavelet methodology
applies a function composed of a complex exponential equation modulated by a Gaussian function for its
fitting procedure, which is generally known as a mother function. One of the famous wavelet mother
functions is the Morlet function [$\psi(t)$] which is given by Eq. 1 (Goupillaud et al., 1984):
$$\psi(t) = \pi^{-1/4} * e^{-\frac{1}{2}t^2} * \left(1 + e^{-f_0^2} - 2e^{-3/4f_0^2}\right)^{-1/2} * \left(e^{if_0 t} - e^{-1/2f_0^2}\right) \tag{1}$$
Where $f_0$ is the center frequency of the wavelet function and $t$ is the time. For the present study, it has been
discretized in n sub-octaves per octave with $n = 50$ for better scale resolution and a wavenumber ($k_0$) of
8. Where the spectral energy of the wavelet is defined as $ABS|\psi(t)|^2$. A logarithmic scaling has been
performed to obtain an adequate highlighting of spectral energy information as follows:
$$\psi(t)'' = \begin{cases} log_{10}(\psi(t) + 1) & \psi(t) \ll 1 \\ log_{10}(\psi(t)) & \psi(t) > 1 \end{cases} \tag{2}$$
$$\psi(t)'' = eps * (\psi(\omega)'') \tag{3}$$
$with\ log_2(eps) = -52.$
It means that eps returns the distance from 1.0 to the next larger double-precision number, that is $2^{-52}$. In
turn, the spectral energy in frequency is presented by performing the temporal integration of the wavelet as
follows:
$$G(f_o) = \int_{\omega_0}^{\omega_{end}} \psi(t)'' \, dt = \int_{-\infty}^{\infty} g(t) * e^{-i2\pi ft} \, dt \qquad (4).$$

## 2.3.    Tsunami propagation model

Although air-ocean interaction has been recognized as a primary mechanism for the global fast-travelling
Tonga 2022 tsunami (Omira et al., 2022), volcano-ocean interaction provided valid explanations for the
near-field observation (Lynett et al., 2022; Pakoksung et al., 2022). As our primary target is Callao Bay,
located approximately 10,000 km away from the HTHH volcano (Figure 3), we only focused on the tsunami
induced by the atmospheric disturbances that followed the volcano explosion. The hypothesis of a point-
source tsunami reaching the South American coast was ruled out considering the modelling results of a
tsunami generated by the underwater explosion of the HTHH volcano (Omira et al., 2022).

**Figure 3. Callao Bay and location of the vessel (Orange vessel) in the offshore port of La Pampilla, Peru at**
**coordinates 11º56' S, 77º11' W.**
To simulate the VMT triggered by the explosion of the HTHH volcano, we used the finite volume GeoClaw
code equipped with atmospheric pressure forcing terms that are handled using the flux-splitting method in
momentum balance (Mandli and Dawson, 2014). This numerical code was validated in various
meteotsunami studies (Kim and Omira, 2021; Kim et al., 2022; Omira et al., 2022). The determination of
atmospheric source terms is based on barometric records from three distinct-selected observations in **B1**,
**B6,** and **B8** (see Table 1 for details). To generate the atmospheric input for the tsunami propagation
modeling, we adopted an approach where we assumed radial symmetry of atmospheric pressure emanating
from HTHH with a constant speed. Specifically, we set the propagation speed at 310 m/s, and interpolated
the atmospheric pressure at each computational grid point using the values from the two closest observed
records. The selection of observed records at Kaitaia (New Zealand), 270001 (Chile), 200006 (Chile), and
Charlotte (US Virgin Islands) was made strategically to capture a diverse range of atmospheric conditions
and geographical locations relevant to our study area. These locations were chosen based on their proximity
to the region of interest and the availability of reliable barometric records. We acknowledge the importance
of validating our atmospheric input against barometric records. We have documented the details of this
validation process in our previous work (Omira et al., 2022), where we discuss the consistency between the
modeled atmospheric input and observed barometric data.
Our simulations covered an area from 170° E to 295° E longitude and from 40° S to 40° N latitude, as
obtained from GEBCO (https://www.gebco.net/). To effectively simulate sea level fluctuations induced by
atmospheric pressure, we used adaptive mesh refinements (AMR) based on both atmospheric pressure and
sea level variations. Six levels of AMR were utilized, beginning with a base resolution of 1° and employing
refinement ratios of 5, 6, 4, 4, and 5 at successive levels. We used resolutions as fine as 1"/40 (~45 meters)
in the areas such as Pampilla, Callao, and La Liberdade, while employing a coarser resolution of up to 1"/2
(~900 meter) in other regions to balance computational efficiency with accuracy. To account for the bottom
friction, GeoClaw software uses Manning's formulation, and the Manning coefficient of 0.02 was
considered in this work. Our simulations were conducted using the Madsen's Boussinesq-type equations,
with a constant value of B = 1/15 (Kim et al., 2017; Madsen and Sørensen, 1992).
The computation was performed on Intel Core i7-8700 CPU 3.2GHz using 10 cores corresponding to 1161
hours of propagation time, and 119 hours of wall time.

### 2.4.    Stresses in moored ships model

Here, we use an analytical model to estimate the loads on a vessel's mooring system under the
hydrodynamic effects of a tsunami (Tahar and Kim, 2003; OCIMF, 2010). The vessel is modelled as a rigid
body with six degrees of freedom (DOFs). The first three DOFs are the surge, sway, and heave ($x$, $y$, and $z$)
of the vessel's centre of gravity ($C_{OG}$), given in the global frame (Figure 4). The other three are the Euler
angles, roll, pitch, and yaw ($\alpha$, $\beta$, and $\gamma$), which describe the local frame rotation status with respect to the
global frame.
We considered two reference frames, the first of which is a global orthogonal inertial frame (fixed) with its
origin located somewhere at the mean water level, where the X-axis points eastward and the Z-axis points
upward for the global frame. The second is an orthogonal non-inertial frame, moving with the vessel, with
its origin located at the $C_{OG}$. The X-axis points toward the bow, while the Z-axis points upward for the local
frame. The roll and pitch initial equilibrium positions are considered at zero degrees.

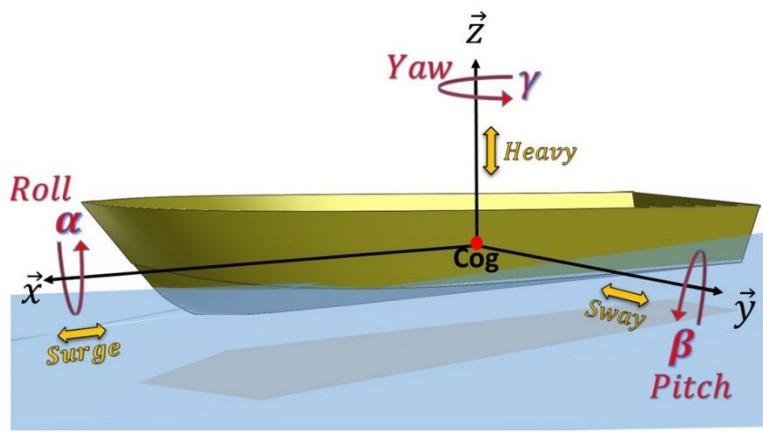


**Figure 4. Definition of Ship Motions in Six Degrees of Freedom, the yellow ones describe translation and the red ones rotation.**

The model contemplates the vessel's hydromechanics and physics characteristics such as hydrostatic stiffness (which considers both buoyancy and gravity forces), vessel mass ($m$), metacentric heights ($GM_{t,l}$), and inertial forces defined by the vessel's moments of inertia around the principal axes ($I_{x,y,z}$) as well as describing the vessel's stability (Journée and Massie, 2001). The vessel's dynamics are described with the equations of a forced and damped mass-spring system which allows the following initial value problem (Eq. 5):

$$\begin{cases} \dot{Y} = f(Y,t) = \begin{bmatrix} \dot{\xi} \\ \ddot{\xi} \end{bmatrix} = \begin{bmatrix} \dot{\xi} \\ M^{-1} \cdot [B \cdot \dot{\xi} + G \cdot (\xi - \xi_0(t)) + F_m(\xi) + F_d(\xi,t)] \end{bmatrix} \\ Y(t=0) = Y_0 \end{cases} \qquad (5)$$

where $Y_0$ is the initial vessel state vector, containing the initial position and the speed of the vessel, $B$ is the damping matrix, $G$ is the stiffness matrix, $\xi$ is defined as the stacking of all the DOFs, $\xi_0$ is the initial equilibrium position, and $F_m$ and $F_d$ are the mooring system forces and the current drag forces (Figure 5).


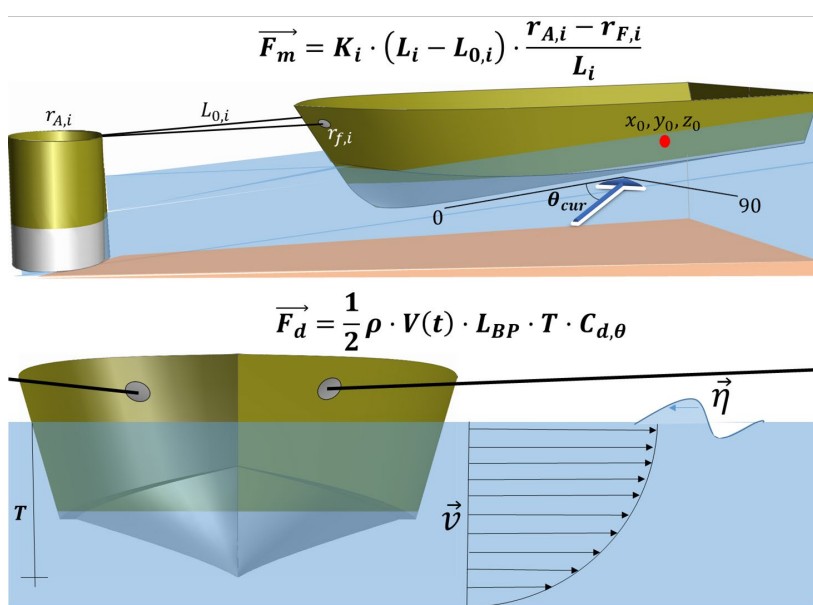


**Figure 5. Schematic layout of Ship Mooring system forces $F_m$ and Drag Forces $F_d$. The dynamics of the moored vessel are defined with a second-order ordinary differential equation (ODE)**

The mooring system $F_m$ is exclusively considered when the movement of the vessel results in a greater
increase in the length of the line $L_i$ compared to the previous position, as shown in the following Eq. (6):
$$F_{m,i} = \begin{cases} K_i \cdot \left(L_i - L_{0,i}\right) \cdot \frac{r_{A,i} - r_{F,i}}{L_i}, & L_i > L_{0,i} \\ 0, & L_i > L_{0,i} \end{cases} \tag{6}$$
Where $K_i$ includes Elastic properties for the i-th line, $L_i$ is the line length, $r_A$ and $r_F$ are the pile and fairlead
positions (where the line is tied to the hull). The Morison drag force model (Oh et al., 2020) is utilized to
calculate the drag force $F_d$. The tsunami current at time $t$ is defined by its speed. The current drag force
changes depending on the angle of incidence, the $F_{d,x,y}$, calculated as Eq (7):
$$F_{d,x,y}(\xi, t) = \frac{1}{2}\rho \cdot V(t) \cdot L_{BP} \cdot T \cdot \begin{pmatrix} \cos(\gamma) & -\sin(\gamma) \\ \sin(\gamma) & \cos(\gamma) \end{pmatrix} \cdot \begin{pmatrix} C_x(|\theta(t)|) \\ C_y(|\theta(t)|) \cdot sgn(\theta(t)) \end{pmatrix} \tag{7}$$
Where $C_x$ and $C_y$ are the drag forces in the $x$ and $y$ axes, $\rho$ is the water density, $V(t)$ is the tsunami current
velocity, $L_{BP}$ is the length between the perpendicular, T is the vessel's draft, and $\gamma$ is the Yaw DOF. We
considers the tsunami current angle $\theta$ to be parallel to the vessel's centerline at zero degrees. Finally, this
first-order ODE is integrated with a Runge-Kutta-4-5 method implemented in the SciPy Python library
(Dormand and Prince, 1980; Shampine, 1986).
The model inputs consist of the vessel's dimensions, hydromechanical and mass properties, the ship's
berthing scheme (including piles and fairleads coordinates), the line properties (such as young modulus and
length), and the temporal tsunami dynamics (including waves and currents time-series) at the vessel's
location. It should be noted that the tsunami dynamics time-series were obtained from the dispersive
Boussinesq-type numerical model, explained in the previous section 2.3, who solve the equations known
as Serre-Green-Naghdi (SGN) instead of the usual shallow water equations (SWE). The SGN equations are
still depth-averaged equations, with the same conserved quantities (Berger and LeVeque, 2023). Their
numerical results can be used to determine the estimation of current velocities due to tsunamis in harbor
environments (Admire et al., 2014; Lynett et al., 2014; Borrero et al., 2015; Ayca et al., 2014). The model
outputs the movements in each DOF and the resulting stress, measured in tons, for each line throughout the
time series.
**3.     Results**
**3.1.     Atmospheric and oceanic data analysis**
The analysis using wavelets is based on the hypothesis that the atmospheric waves' integral characteristics,
such as the periods and propagation velocities, remain reciprocal when resonance is generated between
atmospheric and oceanic waves. This allows the current knowledge extrapolation about atmospheric waves,
which have been extensively studied in the literature, and their effect on oceanic tsunami waves.
The results in Figure 6, Figure 7, and Figure 8 show the analysis of the recorded atmospheric and sea-level
signals, which are composed of three panels for each sensor. The top panel shows the filtered time-series
of both sea level and atmospheric pressure (blue and orange respectively). The orange arrow indicates the
leading Lamb wave arrival time respectively. The lower panel shows the spectral results (wavelets) of the
de-tided series of the ocean surface, the red-blue color scale marks the highest and lowest energy
concentrations respectively; to the right is the Fourier spectrum obtained from integrating the wavelet over
time.
Results in Figure 6 correspond to the tide gauges **NC**, **OW**, and **RI**, DART buoy **D3**, and pressure sensor
**B1** (see Figure 1 for location). Wavelets consistently show four energy groups for sensors tide gauges **NC,**
**OW, RI**, and DART buoy **D3**. These results allows for the identification of the initial characteristics of
tsunami waves forced by atmospheric waves, i.e., the arrival times and wave periods.

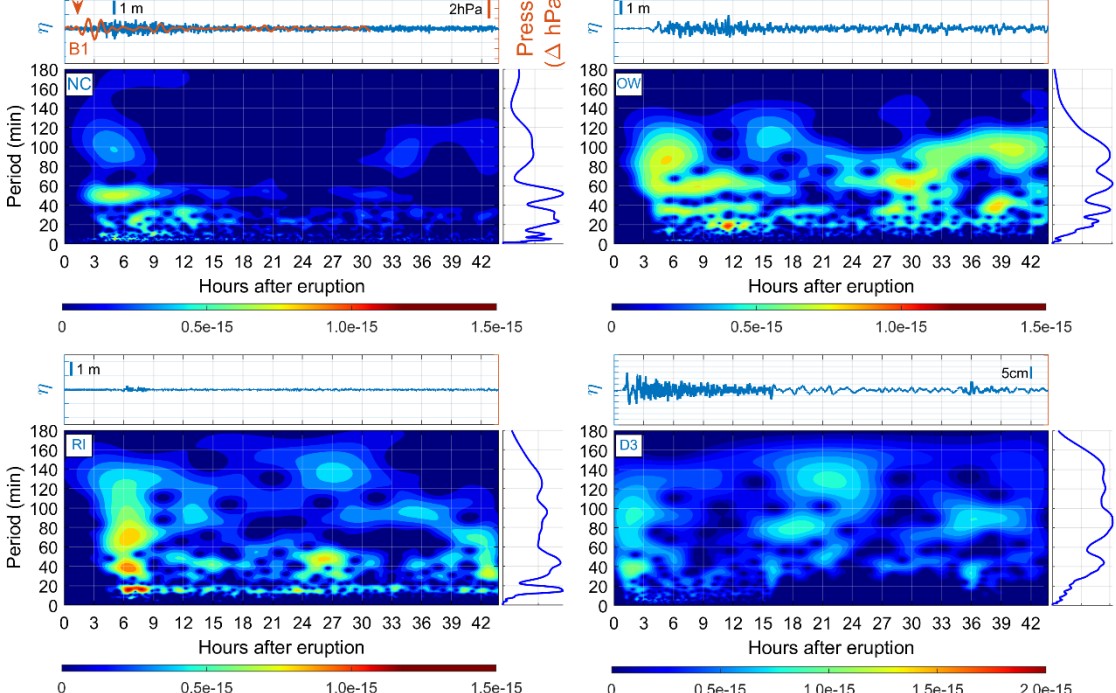


**Figure 6. Analysis of air-pressure and sea-level records at tide gauges NC (North Cape, New Zealand), OW**
**(Owenga, New Zealand) and RI (Rikitea, France), DART buoy D3 (51425, NDBC), and air-pressure sensor B1**
**(North Cape, New Zealand). For each sensor, the time series of the air-pressure sensor (orange line) and tide**
**gauge (blue line) are shown in the upper panel. The orange arrow refers to the arrival time of the leading**
**pressure pulse. The panel to the right of the wavelet is the resulting frequency spectrum (FFT) of the time**
**integration of the wavelet.**
The energy clusters fall within the 5-10 min, 20 to 40 min, 40 to 60 min, and 80-120 min periods, do the
energy clusters suggest that there were multiple mechanisms that generated the Tonga tsunami waves.
These results are consistent with the findings of previous studies (Hu et al., 2023; Kubota et al., 2022;
Omira et al., 2022), where tsunami waves of 1-2 hr. period were also observed in different tidal gauges
when analyzing the similar phenomenon that took place in 1883 during the eruption of the Krakatau
volcano. (Choi et al., 2003; Pelinovsky et al., 2005).
Figure 7 shows far-field results (deep waters near the Peruvian coast) in sensors **D1-B3** and **D2-B5**. Air-
pressure time-series shows the 2 hPa pulse associated with the leading Lamb wave arrived first, followed
by second disturbances travelling at more than 200 m/s (Hu et al., 2022; Omira et al., 2022). DART buoy

wavelets provide more spectral information related to the physical properties of these atmospheric waves coupled in the ocean. For example, DART Buoy **D2** shows that the first pulse coincides with the leading Lamb wave between 30- and 60- minute periods arriving 9 hours after the main eruption. Then, there is a group of energy contained in four ranges of periods: (i) about 10 min, (ii) between 20 and 40 min, (iii) between 60 and 90, and (iv) between 100 and 140 min. The latter is possibly associated with the air-ocean Proudman resonance that occurred on the Tonga Trench and propagated as common tsunami gravity waves towards the Southern American coast (Omira et al., 2022).

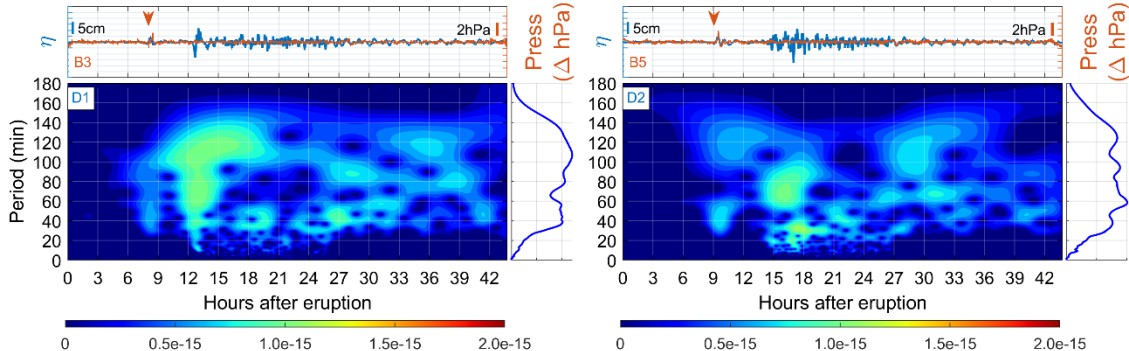

**Figure 7. Analysis of air-pressure and sea-level records at DART buoys D1 (32413, NDBC) and D2 (32401, NDBC) and atmospheric pressure sensors B3 (330031, Chile) and B5 (250005, Chile). For each sensor, the filtered time series of the atmospheric pressure sensor (orange line) and tide gauge (blue line) are shown in the upper panel. The orange arrow refers to the arrival of the leading pressure pulse. The right panel of the wavelet is the frequency spectrum resulting from the time integration of the wavelet.**

Figure 9 shows the spectral results for other locations along the Pacific American coasts. The South Pacific American instruments are **SA-B4** (Chile), **AR-B8** (Chile), **CL-B5** (Peru) and **AL** (Ecuador). The sensors in Central and North America are **ZT** (Mexico) and **AC** (USA). The wavelets' results in Chile, Mexico, and the USA show several energy groups: one in 40 - 60 min periods, another group between 20 - 40 min, and finally, energy in periods less than 10 min. Those wavelets exhibit a pattern similar to that observed in deep water sensors, with a notable difference regarding the absence of periods close to 120 min.

Subsequently, Figure 8 shows the analysis for sensors **LA** (Ecuador), and **CL-B5** (Peru), 15 km from the vessel accident. The black dotted vertical line on the **CL** wavelet refers to the moment when the ship's moorings break and the oil spill occurs, according to the captain of the ship, Mare Doricum. It can be observed in the **CL** wavelet that: (i) the Lamb wave coupled in the ocean (spectral energy between 30 - 60 min periods), (ii) the mooring break moment coincides with the high period spectral energy (max between 110-130 min period). Additionally, the energy within the 100 to 140 min period is present in deep water, (e.g., at DART buoy **D1** in Figure 7) and amplified exclusively in front of the Ecuador and Peru coasts (**LA** and **CL** tide gauges).

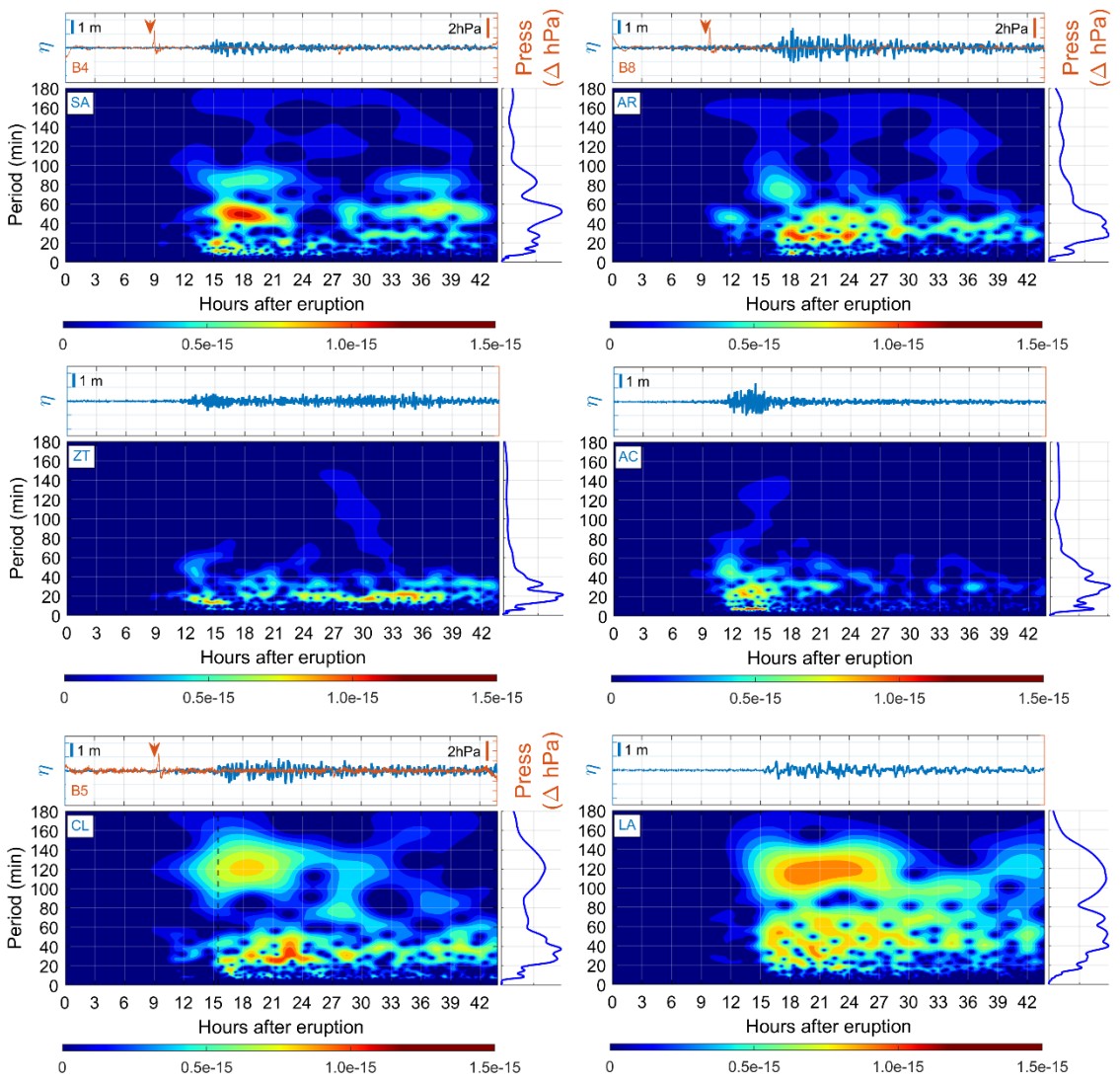

**Figure 8. Analysis of air-pressure and sea level records at the tide gauges of SA (San Antonio, Chile), AR (Arica,**
**Chile), ZT (Zihuatanejo, Mexico), AC (California, USA), CL (Callao, Peru), and LA (La Libertad, Ecuador).**
**The upper panel shows the time series of the atmospheric pressure sensor (orange line) and tide gauge (blue**
**line). The orange arrow refers to the arrival of the leading Lamb wave. For the lower panel, we have the Wavelet.**
**The vertical black dashed line in the CL wavelet refers to the instant when the vessel's moorings broke at La**
**Pampilla port, Peru. The panel to the right of the wavelet is the frequency spectrum.**
**3.2.      Numerical modelling of atmospheric pressure-induced tsunami waves**
The numerical simulations have been calibrated/validated using both far- and near-field instrumental data.
After validation, the significance of tsunami-like waves induced by atmospheric acoustic-gravity waves
and tsunami-induced waves resulting from the submarine explosion were analyzed. We present the results
in Figure 9, which shows the observed and simulated tsunami waveforms near the coast of Peru.
A comparison of observed and simulated tsunami waveforms at DART buoys (**D1** and **D2)** shows that the
Boussinesq numerical model correctly reproduces the first tsunami wave. It also fairly reproduces the
second train of tsunami waves, likely associated with air-ocean resonance in deep ocean (near Tonga
trench), and travelling at common tsunami speeds (Omira et al., 2022; Hu et al., 2023; Kubota et al., 2022).
In shallow waters, the model correctly reproduces the tide gauges **LA** and **CL** (closest to the port of La
Pampilla in Peru.), like in deep waters.
The correlation results obtained from the numerical validation near La Pampilla Port are mainly the
consequence of detailed information such as bathymetry and atmospheric forcers. The low quantity/quality
of the actual measured data nearby the port leads to the results:
• Do not accurately capture perturbations due to nonlinear interactions due to processes such as
refraction, diffraction and reflection.
• Underestimation of the results in the ocean surface wave.
• Underestimation of current velocity.
The wavelets and Fourier transform show a correct trend of the wave characteristics measured in each
sensor, as shown in Figure 9. Wavelets corresponding to both deep and shallow waters present several
groups of waves forced by acoustic waves between 20-40 minutes, periods 40-60 min, and long-period
waves about 120 min (Hu et al., 2023; Kubota et al., 2022; Omira et al., 2022). Likewise, the Fourier spectra
also show two distinguished groups of 40-60 min and 100-120 min in shallow water.

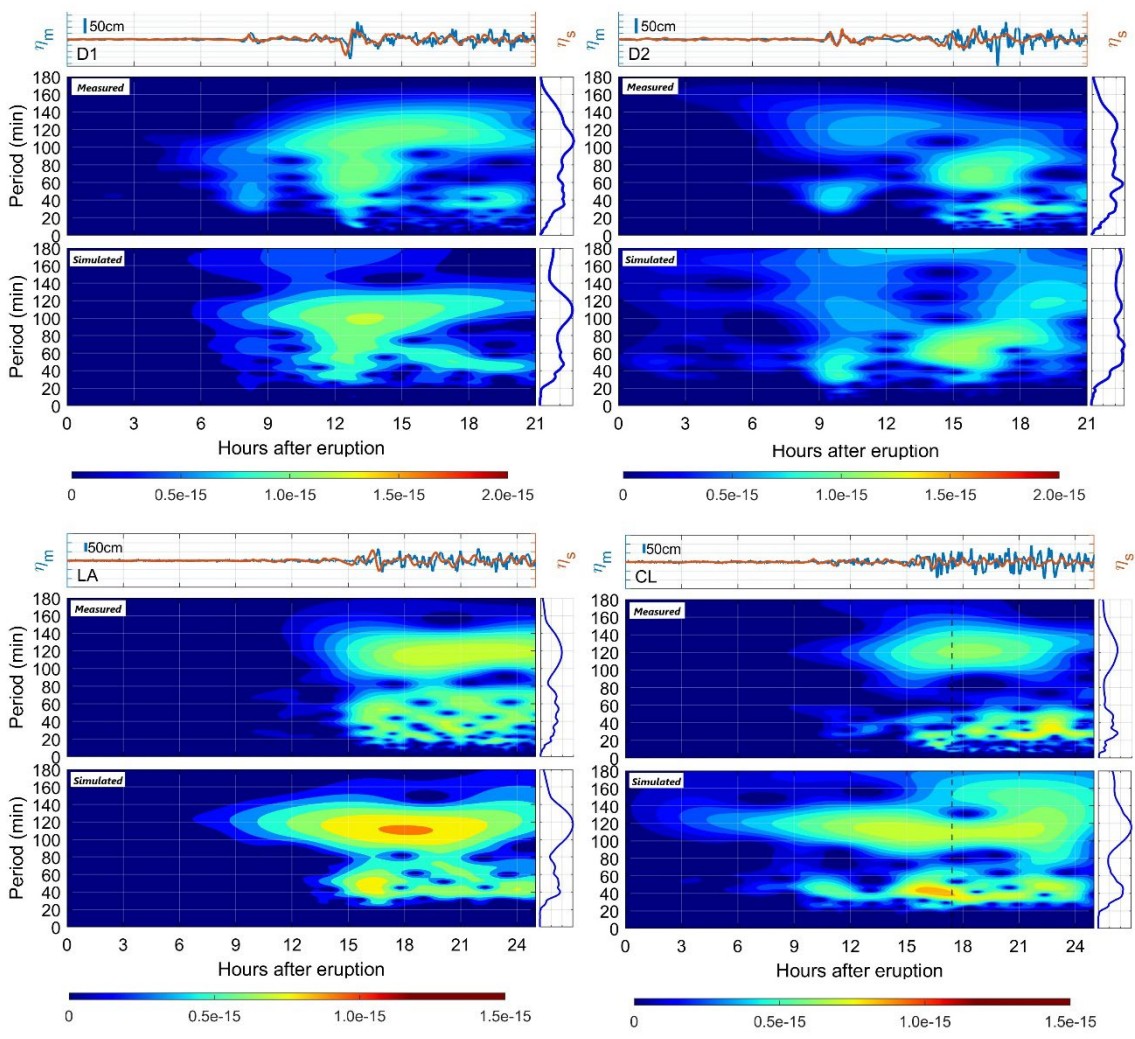

**Figure 9. Validation of numerical results at DART D1 and D2 buoys and tide gauges CL (Callao, Peru), and LA**
**(La Libertad, Ecuador). For each buoy, the measured (blue line) and simulated (orange line) time series are**
**shown in the upper panel. The lower panel shows the measured free sea surface wavelets (upper wavelet) and**
**numerical simulation (lower wavelet). The vertical black dashed line in CL refers to the instant of the vessel**
**mooring break in La Pampilla harbor, Peru. The panel to the right of each wavelet is the frequency spectrum.**
Compared to the DART buoys, the simulations exhibit similar behaviours because they are located at great
depths, where the influence of the bathymetry and the land boundaries is negligible. The simulations
demonstrate that tide gauges reveal variations such as earlier arrival times or more energy, possibly
associated with local effects and limitations in resolution. Despite these limitations, the model is sufficiently
useful for the purpose of the study.
Figure 10 displays simulation results for shallow water tide gauges **GL** and **SA** (located north and south of
Peru, respectively), demonstrating that the 120-minute period wave is exclusive of the coast of Peru and is
likely intensified by local effects. The wavelet analysis indicates that neither the simulated nor the measured
waves show a significant amplification of high-period energy, as seen in the sensors situated off the coast
of Peru.

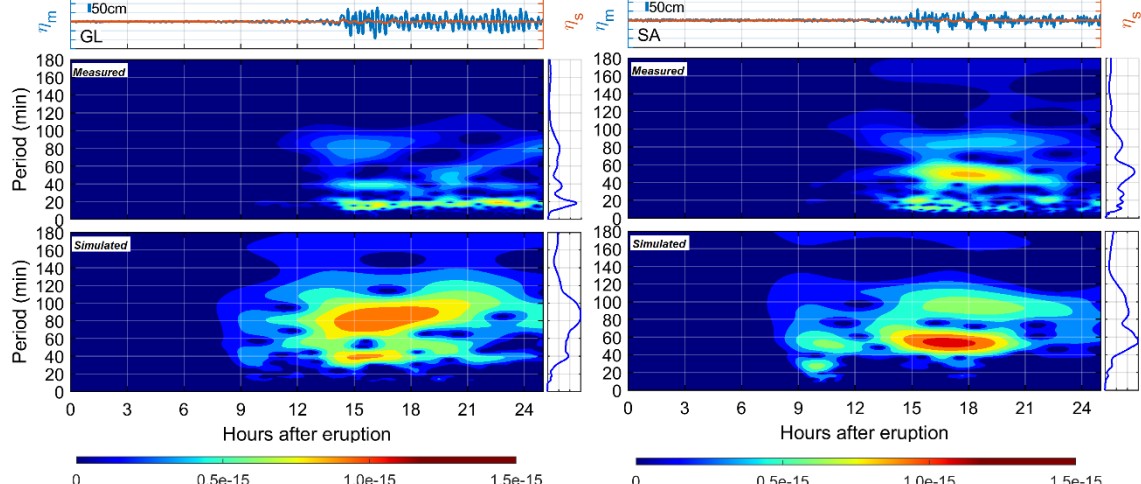


**Figure 10. Comparisons of the numerical results at the GL (Galapagos, Ecuador) and SA (San Antonio, Chile)**
**tide gauges are shown. For each tide gauge, the measured (blue line) and simulated (orange line) time series are**
**shown in the upper panel. The lower panel has measured (upper wavelet) and simulated (lower wavelet) ocean**
**results. The panel to the right of each wavelet is the frequency spectrum.**
**3.3.  Vessel response due to acoustic and tsunami waves**
The model described in Section 2.4 was implemented to estimate the mooring stresses due to tsunami
hydrodynamic effects produced during the Tonga event. The purpose is to demonstrate the variability in
stress levels when exposed to tsunami waves of varying periods and, therefore, different hydrodynamic
conditions.
The current velocity and water elevation time-series were extracted from the Boussinesq tsunami model,
presented in Section 2.3 and validated in Section 3.2 at coordinates -11.932 latitude, -77.181 longitude
(Figure 11.a). The velocity of the current and Sea Water Elevation (SWE) input to the mooring line-loading
model are presented in Figure 11.c. The numerical results suggest that the ocean dynamics over time begins
to be noticeable after 10 hours after the eruption, which coincides with the arrival of the leading Lamb wave
and with maximums at times close to the time of the rupture accident according to the port authority. The
maxima of the current and SWE date values close to 40 cm/s and 0.7 m respectively. On the other hand,
the direction of the current suggests a north-south tilting motion, a similar pattern observed during the entire
simulation.
The mooring scheme (Figure 11.b) is similar to the one found in Terminal 2 of the Port of La Pampilla in
Peru during the mooring-break accident. The vessel is moored to five buoys with eight moorings, one
forward and four at the stern (in addition to two stern anchors anchored at a depth of 18 meters). The line
number six (L6 in Figure 11.b) was the one that broke in the actual accident that occurred at the Port of La
Pampilla 15 hours after the eruption of the THTT volcano.

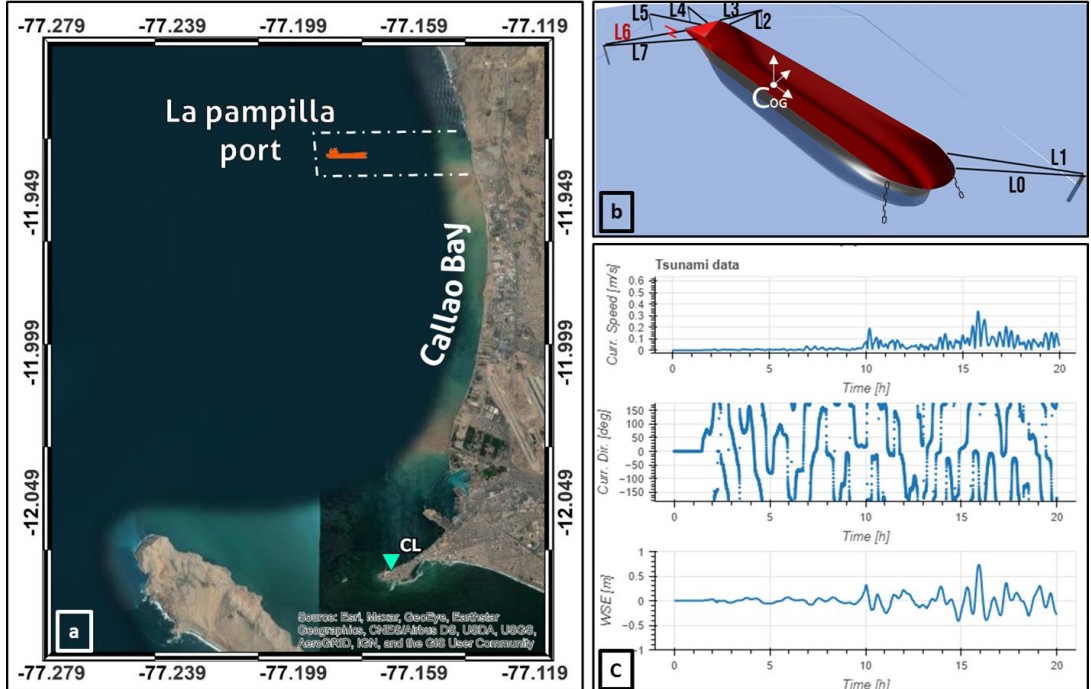


**Figure 11. a, shows the location of the tide gauge CL and the port of La Pampilla at Callao Bay – Peru. b, is the**
**berthing scheme implemented in the mooring lines loads model, where L6 is the line that broke and caused the**
**instability in the mooring system safety. C_OG is the center of gravity of the tanker and origin of the model**
**coordinate system. c, is the time series extracted from the numerical model at tanker location and inputted in**
**the mooring line load model: upper panel is the current velocity, the middle panel is the current direction and**
**the lower panel is the variation of the sea water elevation (SWE).**
In Tables 2 and 3 presents the vessel's description and its hydromechanical characteristics. The entire
mooring system schematic and the data used in the model are provided. The vessel used in this study is not
the oil tanker Mare Doricum, but rather one with comparable physical characteristics. The ship is considered
fully loaded throughout the simulation. (Figure 12).

| LOA | 274 | m |
|------|--------|-----|
| Beam | 48 | m |
| draft | 8 | m |
| Mass | 119.311 | Ton |
| Xcg | 118.3 | m |
| Ycg | 0 | m |

| | | |
|---|---|---|
| Zcg | -6.65 | m |
| Ixx | 21470000 | Ton m2 |
| Iyy | 515780000 | Ton m2 |
| Izz | 515780000 | Ton m2 |
| MGt | 20.042 | m |
| MGl | 471.824 | m |
| Displacement | 100,434 | Ton |
| Waterplane area | 9920.68 | m2 |
| Gz | 99949.56131 | Ton m |
| Groll | 2012895.542 | Ton m |
| Gpitch | 47387108.39 | Ton m |

**Table 2. Description of the vessel used in the mooring stress simulation**

| Line number | Pile position $r_{A,i}$ (m,m,m) | Fairlead position $r_{f,i}$ (m,m,m) | Length $L_{0,i}$ (m) |
|---|---|---|---|
| 0 | (208.27, 80.13, 3.00) | (138.34, 5.65, 6.00) | 103.21 |
| 1 | (208.27, 80.13, 3.00) | (121.00, 13.42, 6.00) | 115.78 |
| 2 | (-152.09, 96.13, 3.00) | (-85.20, 24.30, 6.00) | 105.64 |
| 3 | (-152.09, 96.13, 3.00) | (-92.13, 23.61, 6.00) | 103.54 |
| 4 | (-220.98, 25.94, 3.00) | (-118.19, 3.57, 6.00) | 107.30 |
| 5 | (-225.43, -57.61, 3.00) | (-117.98, -4.43, 6.00) | 125.63 |
| 6 | (-184.75, -120.44, 3.00) | (-89.22, -23.95, 6.00) | 136.73 |
| 7 | (-184.75, -120.44, 3.00) | (-82.04, -24.33, 6.00) | 142.59 |

**Table 3. Description of the mooring system used in the stress simulation**

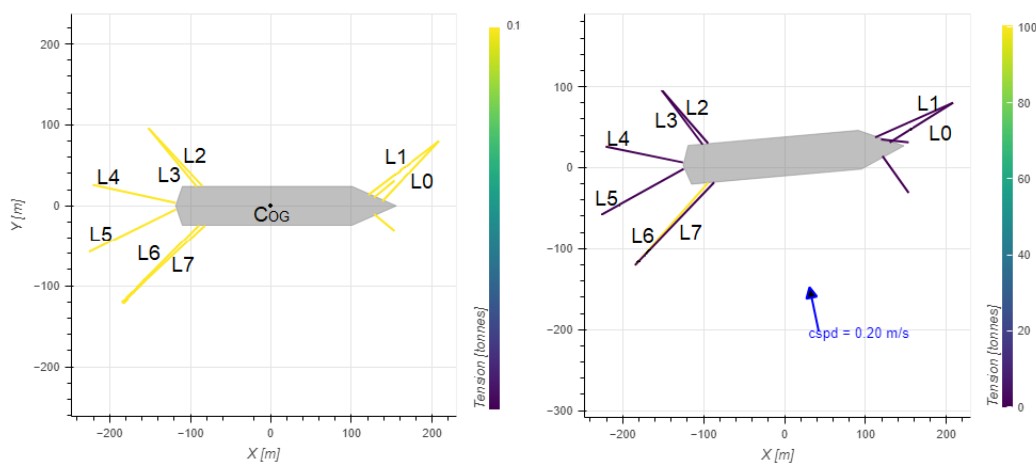

**Figure 12. On the left is the initial layout with the origin at the ship's C_OG and on the right is the layout at the**
**time of mooring breakage, where the yellow and purple colors mark the maximum and minimum stress values,**
**respectively. The cspd value is the current speed at the time of breaking of the mooring line.**
The results of the DOFs along the simulation are presented in Figure 13. The first instances of ship
movement occur about 9 to 10 hours after the volcano erupts, coinciding with the arrival of atmospheric
waves (VMT), with variations in the order of 2 m, 8 m, and 2 degrees for the DOFs Surge, Sway, and Yaw,
respectively, in the movements directly associated with tsunami hydrodynamic loads (López and Iglesias,
2014). Then, 15 hours after the eruption, when the 120-minute period wave is present and the mooring
breaks, further ship motion is generated, drastically increasing the vessel mentioned above DOFs values.
The model results indicate that the movement was caused by the VMT. The anchored ship aligns with the
surge, sway, and yaw; with a maximum deviation of 9 meters, 14 meters, and 5 degrees, respectively, which
is more than enough to produce the breakage of the mooring system according to the port authority of La
Pampilla (CPAAAAE, 2023).

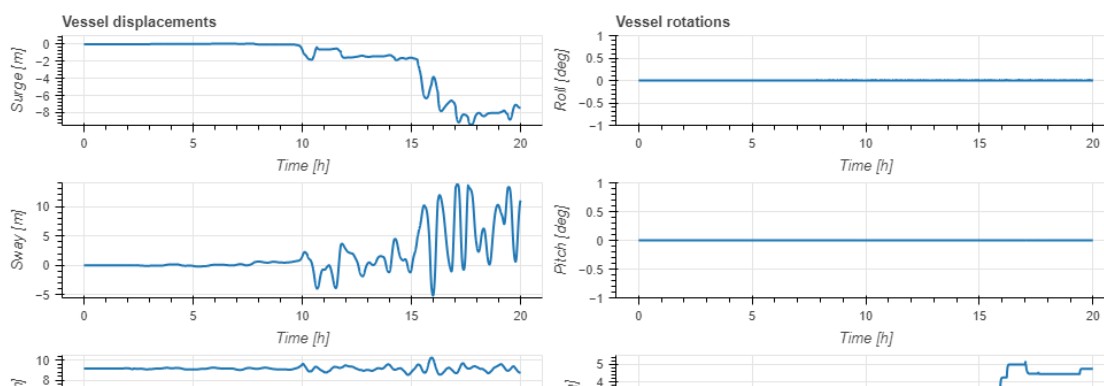

**Figure 13. Time series of ship 6 degrees of freedom obtained from numerical simulation. Measured in hours after the eruption.**

To support the hypothesis that the 120-minute long-period wave along the Peruvian coast caused the
mooring to break, a comparison was made between the stresses obtained by forcing with the VMT time-
series and the tsunami caused only by the submarine explosion. The stress results for each line in both cases
are shown in Figure 14. This illustration shows two things. Firstly, with the VMT, the lines that are primarily
under stress are the starboard moorings lines number 0, 4, and 6, the latter having the maximum load (96
tons), which exceeds the Minimum Breaking Load (MBL) by more than 10 tons. The configuration of the
mooring layout, tsunami wave direction, and hydrodynamic effects can be potential reasons for the increase
in stresses, which could cause the mooring line to break. Secondly, the findings suggest that the VMT
results in a significant increase in mooring stresses, exceeding 10-times the levels observed during the
tsunami-only event (where the VMT is not included in the simulation). These results suggest that the
atmospheric waves generated during the volcanic eruption have triggered a VMT, generating tsunami-like
waves that may have affected mooring safety of vessels berthed in offshore ports in the far field.

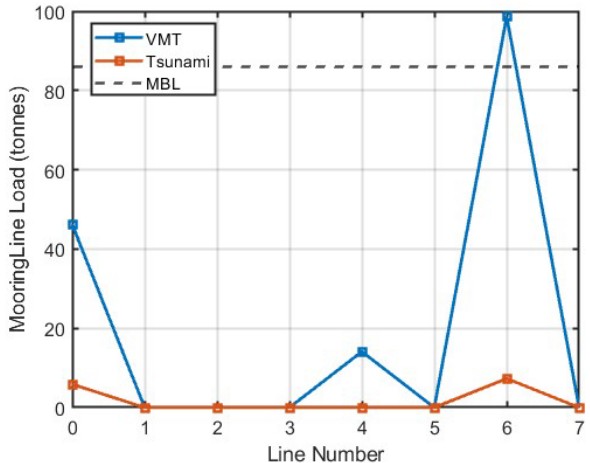

**Figure 14. Maximum stresses obtained from the simulation at each mooring. Blue and orange lines represent the results of the simulations with and without atmospheric waves, respectively.**

## 4.    Conclusions

The propagation of atmospheric waves and their coupling with the ocean were extensively studied following the Tonga event. Although epistemic uncertainties associated with the event are important, it was possible to understand the main drivers and effects of the volcano-meteorological tsunami (VMT) in the mooring safety of vessels moored in offshore ports in the far field. The air-ocean Proudman resonance in deep water was the driving mechanism that caused long period gravitational waves (which had very similar characteristics to those of a tectonic-source tsunami.) to travel along the Pacific Ocean and affect the offshore port La Pampilla in the far field.

The potential of the explosion-induced atmospheric waves to magnify tsunamis once they pass over deep ocean regions has been evidenced. The second train of tsunami-like waves that arrived in the Peruvian coast and affected the mooring safety were generated due to Proudman resonance between acoustic-gravity waves with ocean-gravity waves, which produced an air–ocean energy transfer that led to an increase in tsunami energy in the far field instead of losing it. This happened near the Tonga trench in the propagation direction of South America (Figure 3-c in Omira et al., 2022).

The presence of the high period waves exclusively in the tide gauges of Peru and Ecuador (**CL** and **LA,** respectively) could be explained by several processes associated with shoaling. These processes could include the width and slope of the continental shelf (which is wider on the Peruvian coast than on the Chilean coast, for example) and the effects associated with topographic boundaries and their geometry, such as the natural bay oscillation modes.

The spectral analysis results establish the influence of the atmospheric waves generated by the HTHH volcano. Considering that, the 120-minutes long-period waves were associated with the air-ocean-resonance of the Tonga event and that numerical simulations additionally show the mooring line stress using the VMT time-series, it is possible to conclude that the Tonga tsunami caused the overstressing and subsequent accident in the port of La Pampilla, Peru.

Hydrodynamic loads on the vessel's hull due to tsunamigenic phenomena can threaten the stability of moored vessels. These loads are mainly due to drag forces (driven by the tsunami kinetic energy) that affect ship stability. Based on the ship DOF, the results suggest that VMT affect mainly the horizontal plane motions, two associated with displacements (sway and surge) and the other with the angle of rotation about its vertical axis (yaw). This is likely explained by the large amount of kinetic energy that tsunamis have in their propagation, which travel at high speed, but with little wave elevation. The VMT produced during the Tonga 2022 event was accompanied by long-period waves and currents, which could affect the stability of the mooring system in the port of La Pampilla, Peru.

The hydrodynamic effects of very long period tsunami waves can generate damage similar to those of tsunamis of tectonic origin, affecting elements such as infrastructure, vessels, merchandise and people in port environments.

## 4.1. Final Considerations

Coastal and port infrastructure are not prepared to respond preventively to these Tonga-type tsunamis, leaving them "unprotected", as tsunami warnings are not issued once a volcano eruption is known. Furthermore, in the analyzed event, the initial ocean disturbance arrived earlier than anticipated because the atmospheric waves produced a VMT that travelled at sonic velocity. This statement holds relevance as state and international authorities are responsible for maritime safety and the creation of cautions-warnings and suggestions to aid distinct users in coastal and offshore locations.

This event showed the need for Tsunami Early Warning Systems (TWS) to be prepared to include atmospheric waves and detect them from existing monitoring sensors. In addition, Standard Operational Procedures need to include protocols for these events to avoid damage to port facilities and ships, such as the breaking of moorings and ship collisions. These events can also generate local flooding increasingly far away from the origin and affect the population and coastal infrastructures (cities, nuclear reactors, petrochemical industry, etc.). Therefore, efforts towards the incorporation of tsunamis caused by volcanic acoustic waves in tsunami warning systems are needed.

## 5. Author contribution

Supervision and methodology (G.M and A-Q.I); Waves simulations (K.J and O.R); revision (B.M and O.R). All authors have read and agreed to the published version of the manuscript.

## 6. Competing interests

At least one of the co-authors is a member of the editorial board of Natural Hazards and Earth System Sciences.

**7.    Acknowledgement**
This research was supported by an FPU (Formación de Profesorado Universitario) grant from the Spanish
Ministry of Science and Innovation (MCINN) to the first author. We have gratitude to The Ocean Energy
and Offshore Engineering Group of IHCANTABRIA for the model of mooring stresses. In addition, the
authors of this work would like to thank the various state institutions that have provided measured
atmospheric data mentioned in Chapter 2: IDEAM (Colombia), SENAMHI (Perú), DGAC (Chile), NIWA
(New Zealand), NOAA (USA) and IOC.

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
