# Peer review of "safety: The 2022 Tonga event"

_EGUsphere, 2024_

## Referee Comment (RC2)

2022.

[referee-annotated manuscript omitted]

---

## Author Response (AR1)

Referee comments are in black Calibri

The paragraphs of the document in question will be in black Calibri in quotation marks ""

*Our replies to reviewers are comments in blue and italics.*

> Paragraphs that are part of the document will be written in blue, indented and new text highlighted in yellow.

**Referee 1**

**COMMENT Nº 1:**

I just would like to note that the dominant period of 120 minutes obtained by the authors was also observed when the similar phenomenon, that took place in 1883 during the eruption of the Krakatau volcano, was being analyzed (Choi et al Simulation of the trans-oceanic tsunami propagation due to the 1883 Krakatau volcanic eruption. Natural Hazards and Earth System Sciences, 2003, vol. 3, 321 – 332; Pelinovsky et al. Analysis of tide-gauge records of the 1883 Krakatau tsunami. Tsunamis: case studies and recent developments, Advances in Natural and Technological Hazards Research, vol. 23, Springer, 2005, 57-77), so it would be worth mentioning in the paper.

**RESPONSE TO COMMENT Nº1:**

*First of all, thank you again for your comments and bibliographic recommendations regarding the Krakatoa event of 1883, they are quite accurate and help to give more support to the results, in addition to improving the understanding of this type of tsunamis of volcanic origin. We have made the inclusions in as follows:*

> The energy clusters fall within the 5-10 min, 20 to 40 min, 40 to 60 min, and 80-120 min periods, do the energy clusters suggest that there were multiple mechanisms that generated the Tonga tsunami waves. These results are consistent with the findings of previous studies (Hu et al., 2023; Kubota et al., 2022; Omira et al., 2022), where tsunami waves of 1-2 hr. period were also observed in different tidal gauges when analyzing the similar phenomenon that took place in 1883 during the eruption of the Krakatau volcano. (Choi et al., 2003; Pelinovsky et al., 2005).

**COMMENT Nº 2:**

I would like to point out minor inaccuracies in References (not completed data):

"Ohgaki, K., Yoneyama, H., Suzuki, T., and IEEE: Evaluation on safety of moored ships and mooring systems for a tsunami attack, 2008"

"Tahar, A. and Kim, M. H.: Hull/mooring/riser coupled dynamic analysis and sensitivity study of a tanker based FPSO, https://doi.org/10.1016/j.apor.2003.02.001, 2004"

**RESPONSE TO COMMENT Nº2:**

*Thank you very much for making us realize that these references are incomplete. The new ones added in the bibliography section are:*

> Ohgaki, K., Yoneyama, H., and Suzuki, T.: Evaluation on Safety of Moored Ships and Mooring Systems for a Tsunami Attack, in: OCEANS 2008 - MTS/IEEE Kobe Techno-Ocean, 1–6, https://doi.org/10.1109/OCEANSKOBE.2008.4530986, 2008.

> Tahar, A. and Kim, M. H.: Hull/mooring/riser coupled dynamic analysis and sensitivity study of a tanker-based FPSO, Appl. Ocean Res., 25, 367–382, https://doi.org/10.1016/j.apor.2003.02.001, 2003.

**Referee 2**

**COMMENT Nº 1:**

Some of the sentences in the abstract are unclear, making it difficult to apprehend the overall study and major findings from the abstract. For example, The authors mention two vessels, one docked at La Pampilla Port on the day of the Tonga event and the other to which the mooring system model applied. They mention a 'similarity' between these two vessels, **but the remark (here) that the authors would like to point out is not clear.** On the other hand, one of the main findings given in the abstract is, "The results suggest that the Tonga tsunami event could be responsible for the movement and loss of positioning of the vessel." One can already expect that considering the timing of the incident; **so the authors should emphasize the results/outputs of the study supporting this statement**. It is important to summarize those in the abstract. Another finding, "Furthermore, atmospheric waves significantly increased mooring stresses, particularly on the starboard quarter moorings." **should also be rewritten more precisely, including the answers to some questions like**:

**RESPONSE TO COMMENT Nº1:**

*Thank you very much for your appreciation, we think it is important to clarify this sentence.*

*What we want here is to make a distinction between the ship used for the simulation and the real one. We have worked with a ship with physical and inertial characteristics very similar to the real one (this is because we did not have precise information about the conditions of the real ship. We considered that this distinction should be made in order to give transparency to the results and to focus on the fact that the exercise is exclusively for research purposes.*

- is it any characteristics of atmospheric waves that increased the mooring stresses or
- is there any other mechanism serving to that increase?
- How much is that 'significant' increase and
- is it compared to which condition?

*Thank you very much for pointing all this out. We have decided write a new one:*

**Abstract.** The explosion of the Hunga Tonga-Hunga Ha'apai volcano on January 15, 2022 (Tonga22) was the origin of a volcano-meteorological tsunami (VMT) recorded worldwide. At a distance exceeding 10,000 kilometres from the volcano and 15 hours after its eruption, the moorings of a ship in the Port of La Pampilla, Callao (Peru), failed, releasing over 11,000 barrels of crude oil.

This study delves into the profound implications of the Tonga 22 event, investigating whether it could have led to the break of the mooring system. We conducted a comprehensive analysis of this significant event, examining the frequency content of the time series recorded at tide gauges, DART buoys, and barometers in the Southern Pacific Ocean. Our findings revealed that the maximum energy of the spectra corresponds to 120 minutes of wave period off the coast of Peru, with the arrival time of these waves coinciding with the time of the accident in the Port.

We used a Boussinesq model to simulate the propagation of the Volcano-Meteorological tsunami from the source to the Port in Peru to study the impact of those waves on the mooring system. We used the synthetic tsunami recorded in the Port as input for the model that simulates mooring line loads based on the ship's degrees of freedom. The results suggest that the 120-minute wave triggered by the VMT could significantly increase mooring stresses due to the resulting hydrodynamic effects, exceeding the Minimum Break Load (MBL).

We conclude that the propagation of the long wave period generated by the VMT caused overstresses in moored lines that triggered accidents in port environments. This event showed the need to prepare Tsunami Early Warning Systems and port authorities for detecting and

managing VMTs induced by atmospheric acoustic waves. The work provides new insights into the far-reaching impacts of the Tonga 2022 tsunami.

**INTRODUCTION**

**COMMENT Nº 2:**

The author suggests somewhat linking paragraphs and sentences within the paragraphs for fluent reading.

**RESPONSE TO COMMENT Nº2:**

*You are right, thank you very much. We have taken this into account to improve the readability of the document.*

**COMMENT Nº 3:**

Figure 1: The reviewer highlights the need to help the reader: identification of oceanic and land features mentioned in the text; indicating the names of the important (surrounding) geographical objects on the map.

**RESPONSE TO COMMENT Nº3:**

[Figure]

Figure 1. a, locations of the measuring instruments. Red circles show the atmospheric pressure sensors, the light blue triangles show the tide gauges, and the yellow circles refer to the Dart Buoys. b, shows the HTHH volcano (orange triangle) and Tonga trench location. c, is the Callao bay zoom, the orange vessel represents the location of terminal 2 of the port of La Pampilla, Peru.

**COMMENT Nº 4:**

Lines 56-58: This statement needs to be checked. The reviewer agrees that the VMT traveled at faster speeds, had a global reach, and somewhat affected the far-field coasts but did not cause noticeable damages or human fatalities since the observed amplitudes due to VMT were only in the order of 20-50

cm. Please clarify or support further with additional information and numbers (like the amplitudes or give specific examples/locations, etc.)

"The Tonga VMT was exceptional as it travelled at faster speeds than common tsunamis, had a global reach, affected the far-field coasts, and caused noticeable damages, human fatalities, and coastal hydrodynamic effects (Terry et al., 2022; Omira et al., 2022; Lynett et al., 2022)".

**RESPONSE TO COMMENT Nº4:**

*Thank you very much for your comment: throughout the bibliography of the studies of this event, the different damages and deaths generated in the Pacific coasts have been documented. In the mentioned references. We do not go into detail on the different damages because we wanted to make the reading lighter. There are also several examples that support this sentence:*

*In Kochi prefecture 5 boats were sunk and capsized in the port of Sakihama, (https://oceancrew.org/news/more-than-30-boats-sank-due-to-the-tsunami-in-japan_17-01-2022/). In peru two people died due to this event (https:// www. theguardian. com/ world/ 2022/ jan/ 17/ two- drown- in- peru- as- abnor mally- big- waves- from- tonga- volca no- hit- coast). Also strong currents in ports of Mexico were reported as a direct effects of the tsunami, and the largest tsunami heights (2 m) were measured by tide gauges on the Manzanillo-Colima and Ensenada-Baja California coast, where strong currents were also observed" (Ramírez-Herrera et al., 2022).*

*We have included some examples in the document as you propose:*

The Tonga VMT was exceptional as it travelled at faster speeds than common tsunamis, had a global reach, affected the far-field coasts, and caused noticeable damages, human fatalities, and coastal hydrodynamic effects (Terry et al., 2022; Omira et al., 2022; Lynett et al., 2022). For instance in Kochi prefecture 5 boats were sunk and capsized in the port of Sakihama in the city of Muroto (https://oceancrew.org/news/more-than-30-boats-sank-due-to-the-tsunami-in-japan_17-01-2022/), two people died in Peru (INDECI, 2024) and strong currents in ports of Mexico were reported as a direct effects of the tsunami (Ramírez-Herrera et al., 2022).

INDECI: https://www.gob.pe/institucion/indeci/noticias/576687-inician-acciones-de-respuesta-luego-de-oleajes-en-el-litoral, last access: 22 April 2024.

**COMMENT Nº 5:**

Lines 69-70: It is not clear what the 'previous ones' here refer to. The authors can simply use 'similarly' to avoid confusion, if applicable. Furthermore, regarding the statement, it would be useful to explain those incidents since they are directly related with the concept of the proposed study.

''This also applies to tsunamis induced by atmospheric disturbances, which, like the previous ones, have caused damage to ships, moored vessels, and small bays (Imamura et al., 2022; Thomson et al., 2009)",

**RESPONSE TO COMMENT Nº5:**

*Thank you for improving the wording in this section: we will change "previous ones" to "similarly" as you suggest. We did not include the accidents description because we assume that making the citation is enough for support the sentence (and make the reading lighter), but you are right, it would be useful include at least one:*

This also applies to tsunamis induced by atmospheric disturbances, which, similarly, may have caused damage to ships, moored vessels, and small bays (Imamura et al., 2022; Thomson et al., 2009). It has been observed in these studies that strong currents, often accompanying long-period waves, increase the probability of generating large catastrophes in harbours(Sakakibara et al., 2010; Zheng et al., 2022; Lynett et al., 2012). For instance (Shigeki and Masayoshi, 2009)

studied numerically the mooring loads due to large-scaled tsunamis in GNL-carriers vessels and found out that the drag forces due to the currents induce large sway and surge motions are strongly important.

**COMMENT Nº 6:**

Lines 74-75: The motions in the vessel's horizontal plane are misnamed here. They should be sway, surge and yaw in the horizontal plane and heave, pitch and roll in the vertical plane!

"Authors such as López and Iglesias (2014) agree with the hypothesis that the motions in the vessel's horizontal plane (named roll, heave, and yaw) "

**RESPONSE TO COMMENT Nº6:**

*Thanks referee for highlighting this very clear mistake. Will be changed in the document by: sway, surge and yaw.*

"Authors such as López and Iglesias (2014) agree with the hypothesis that the motions in the vessel's horizontal plane (named sway, surge and yaw) "

**COMMENT Nº 7:**

Figure 3: It would be helpful to the reader if the authors could show/write/name Callao Bay and La Pampilla Port on the map here and provide coordinates for La Pampilla Port, preferably in the figure caption.

**RESPONSE TO COMMENT Nº7:**

[Figure]

**Figure 3. Callao Bay and location of the vessel (Orange vessel) in the offshore port of La Pampilla, Peru at coordinates 11º56' S, 77º11' W.**

**2. DATA AND METHODS**

**2.2 SPECTRAL ANALYSIS**

**COMMENT Nº 8:**

The reviewer strongly suggests explaining (in detail) the methodology that the authors applied in this analysis.

**RESPONSE TO COMMENT Nº8:**

*Thank you very much for your appreciation. A more detailed definition has not been made in this section because it is a methodology quite used in the literature and it was not considered relevant to go deeper, however, more detail will be included in the method used in the spectral section as you suggest:*

Spectral analysis is a practical tool for identifying the characteristic frequencies and energy levels in a time series, particularly when it is composed of the superposition of signals with different frequencies. This method has found extensive application in the study of tsunamis (Abe, 2011; Rabinovich, 1997; Shevchenko et al., 2011; Satake et al., 2013; Baptista et al., 2016; Xu et al., 2022). In the time domain, we use wavelets to perform spectral analysis, allowing the estimation of the evolution of the spectral energy and, consequently, the first instants of energy increase for a given specific period. Wavelet methodology applies a function composed of a complex exponential equation modulated by a Gaussian function for its fitting procedure, which is generally known as a mother function. One of the famous wavelet mother functions is the Morlet function [$\psi(t)$] which is given by Eq. 1 (Goupillaud et al., 1984):

$$\psi(t) = \pi^{-1/4} * e^{-\frac{1}{2}t^2} * \left(1 + e^{-f_0^2} - 2e^{-3/4 f_0^2}\right)^{-1/2} * \left(e^{if_0 t} - e^{-1/2 f_0^2}\right) \qquad (1)$$

Where $f_0$ is the center frequency of the wavelet function and $t$ is the time. For the present study, it has been discretized in n sub-octaves per octave with $n = 50$ for better scale resolution and a wavenumber ($k_0$) of 8. Where the spectral energy of the wavelet is defined as $ABS|\psi(t)|^2$. A logarithmic scaling has been performed to obtain an adequate highlighting of spectral energy information as follows:

$$\psi(t)'' = \begin{cases} log_{10}(\psi(t) + 1) & \psi(t) \ll 1 \\ log_{10}(\psi(t)) & \psi(t) > 1 \end{cases} \qquad (2)$$

$$\psi(t)'' = eps * (\psi(\omega)'') \\ with \ log_2(eps) = -52. \qquad (3)$$

It means that eps returns the distance from 1.0 to the next larger double-precision number, that is, $2^{-52}$. In turn, the spectral energy in frequency is presented by performing the temporal integration of the wavelet as follows:

$$G(f_o) = \int_{\omega_0}^{\omega_{end}} \psi(t)'' \, dt = \int_{-\infty}^{\infty} g(t) * e^{-i2\pi f t} \, dt \qquad (4)$$

**2. 3 TSUNAMI PROPAGATION MODEL**

**COMMENT Nº 9:**

Lines 144-152: As the outputs of the study strongly depend on the results of tsunami propagation modeling, this part certainly needs to be improved. The methodology used for producing atmospheric input should be explicitly provided, as should the produced signal (preferably compared with barometric records).

**RESPONSE TO COMMENT Nº9:**

*Thank you for your feedback: the following text has been included in the manuscript:*

To generate the atmospheric input for the tsunami propagation modeling, we adopted an approach where we assumed radial symmetry of atmospheric pressure emanating from HTHH with a constant speed. Specifically, we set the propagation speed at 310 m/s, and interpolated

the atmospheric pressure at each computational grid point using the values from the two closest observed records. The selection of observed records at Kaitaia (New Zealand), 270001 (Chile), 200006 (Chile), and Charlotte (US Virgin Islands) was made strategically to capture a diverse range of atmospheric conditions and geographical locations relevant to our study area. These locations were chosen based on their proximity to the region of interest and the availability of reliable barometric records. We acknowledge the importance of validating our atmospheric input against barometric records. We have documented the details of this validation process in our previous work (Omira et al., 2022), where we discuss the consistency between the modeled atmospheric input and observed barometric data.

**COMMENT Nº 10:**

Lines 153-160: The reviewer would like to see the computational domains (their extents) employed in the study on a map, with corresponding resolutions in meters as well.

*Thank you very much for your comment. We have not included it because we did not see much usability based on the objectives and scope of the work, but it is true that providing information associated with numerical meshes is very accurate. The following text has been included in the manuscript:*

**RESPONSE TO COMMENT Nº10:**

Our simulations covered an area from 170° E to 295° E longitude and from 40° S to 40° N latitude. To effectively simulate sea level fluctuations induced by atmospheric pressure, we used adaptive mesh refinements (AMR) based on both atmospheric pressure and sea level variations. Six levels of AMR were utilized, beginning with a base resolution of 1° and employing refinement ratios of 5, 6, 4, 4, and 5 at successive levels. We used resolutions as fine as 1''/40 (~45 meters) in the areas such as Pampilla, Callao, and La Liberdade, while employing a coarser resolution of up to 1''/2 (~900 meter) in other regions to balance computational efficiency with accuracy.

**2.4 STRESSES IN MOORED SHIPS MODEL**

**COMMENT Nº 11:**

Figure 5: The reviewer suggests writing the equations in bigger font size and a darker color for visibility.

**RESPONSE TO COMMENT Nº11:**

*Thank you very much for your comment. Figure 5 has been modified as follows.*

[Figure]

**Figure 5. Schematic layout of Ship Mooring system forces $FF_{mm}$ and Drag Forces $FF_{dd}$. The dynamics of the moored vessel are defined with a second-order ordinary differential equation (ODE)**

**COMMENT Nº 12:**

As the tsunami current velocity is one of the inputs in the moored ships model as a time series, it is important to explain how it is handled in the vertical dimension and solved in the numerical model.

**RESPONSE TO COMMENT Nº12:**

*This is an accurate assessment. Unfortunately, it was not possible to obtain in situ current data (not even at a nearby location) at the time of the accident, due to the lack of availability, so we have used the results of the model (validated with the ocean surface elevation) in the vessels location. The time series were obtained from the dispersive Boussinesq-type numerical GeoClaw model, who solve the equations known as Serre-Green-Naghdi (SGN) instead of the usual shallow water equations (SWE). The SGN equations are still depth-averaged equations, with the same conserved quantities (Berger and LeVeque, 2023). The advantage of using Boussinesq-type mode is that it allows to give a better estimation of the current velocity compared to a SWE, although it generally tends to underestimate the speed peaks (Lynett et al., 2013). Their numerical results can be used to determine the estimation of current velocities due to tsunamis in harbor environments (Admire et al., 2014; Lynett et al., 2014; Borrero et al., 2015; Ayca et al., 2014).*

*The explanation of the origin of the time series of the dynamics generated by the tsunami (input to the model of loads on the ship's moorings) has been included:*

The model inputs consist of the vessel's dimensions, hydromechanical and mass properties, the ship's berthing scheme (including piles and fairleads coordinates), the line properties (such as young modulus and length), and the temporal tsunami dynamics (including waves and currents time-series) at the vessel's location. It should be noted that the tsunami dynamics time-series were obtained from the dispersive Boussinesq-type numerical model, explained in the previous section 2.3, who solve the equations known as Serre-Green-Naghdi (SGN) instead of the usual shallow water equations (SWE). The SGN equations are still depth-averaged equations, with the same conserved quantities (Berger and LeVeque, 2023). Their numerical results can be used to determine the estimation of current velocities due to tsunamis in harbor environments (Admire et al., 2014; Lynett et al., 2014; Borrero et al., 2015; Ayca et al., 2014). The model outputs the movements in each DOF and the resulting stress, measured in tons, for each line throughout the time series.

Berger, M. J. and LeVeque, R. J.: Implicit Adaptive Mesh Refinement for Dispersive Tsunami Propagation, 2023.

**RESULTS**

**COMMENT Nº 13:**

Lines 241-242: What is the rationale behind the selection of 310 m/s for the traveling speed of atmospheric waves, while the authors here give an estimation of 324 m/s? The reviewer also recommends a discussion of the significance of this speed selection as it is the primary factor for potential Proudman resonance effects.

"The manometer station B1 and tide gauge NC allow the estimation of the propagation velocities of the atmospheric disturbances as approximately 324 m/s."

**RESPONSE TO COMMENT Nº13:**

*This value was estimated pointwise in a near-field barometer (B1, located in New Zealand), and is directly associated with the first arrival of the atmospheric wave, i.e., the leading Lamb wave. Some authors such as Hu et al. (2023) give estimates in the range of 315 a 340 m/s . Omira et al. 2022*

*concluded that the optimum velocity value of 322 m s-1 shows the best agreement between simulations and observations.*

*It should be clarified that the Proudman resonance has occurred between the acoustic-gravitational wave and oceanic long waves in very deep water (such as the Tonga trench) and has been an amplifying factor of the second tsunami wave train (Omira et al., 2022).*

**COMMENT Nº 14:**

The reviewer thinks that the calibration/validation of the tsunami modeling results is significant, especially at the locations near the La Pampilla port, at LA and CL. Therefore, the discrepancies between the observed and simulated waveforms, especially time histories (which do not seem negligible as claimed) and their effects on the ship moored model need to be further discussed.

**RESPONSE TO COMMENT Nº14:**

*For validation, it is important to understand the influence of instrument location and the resolution of the detail grid (~450 m). As mentioned in the document, to measure waves in the ocean we have two types. Dart buoys (deep water) and tide gauges (on the coast).*

*What happens with DART buoys, for example, **D1 (in below Figure 13.1)**, is that, being in deep water, the waves in their propagation suffer less processes because there is no reflection, and just a little refraction and diffraction, so the model solves it better. Another similar case is the tide gauge **LA (in below figure)**, which, despite being on the coast (Figure. 13.1), does not present major obstacles in the propagation process, so the results in this tide gauge are admissible.*

[Figure]

*Figure comment 14.1. Validation of numerical results at DART **D1** and tide gauges **LA** (La Libertad, Ecuador). For each buoy, the measured (blue line) and simulated (orange line) time series are shown in the upper panel. The lower panel shows the measured free sea surface wavelets (upper wavelet) and numerical simulation (lower wavelet). The panel to the right of each wavelet is the frequency spectrum.*

[Figure]

*Figure comment 14.2. Location of the tide gauge **La**, in La Libertad – Ecuador.*

*Now, what happens with the **CL** tide gauge is that it is located inside the port of Callao (**Figure 13.3**), so it captures the initial disturbances well, but not so much the disturbances within the port. It has not been possible to obtain high-resolution port-scale bathymetry (<10 m resolution) to improve the numerical results.*

[Figure]

*Figure comment 14.3. Location of the tide gauge **CL**, in Callao Port – Peru.*

*Despite these limitations, the model is sufficiently useful for the exercise. In deep water and in LA, the model does better and thus can be assumed to perform well in the ship's location: an offshore port 4.5 km away from shore and 18 m deep. This is validated by the results of the ship model, where we obtain the increases in mooring stresses coincident with what happened*

*It is true that more discussion of this is needed in the document, so the following will be included:*

In shallow waters, the model correctly reproduces the tide gauges LA and CL (closest to the port of La Pampilla in Peru.), like in deep waters. The correlation results obtained from the numerical validation near La Pampilla Port are mainly the consequence of detailed information such as bathymetry and atmospheric forcers. The low quantity/quality of the actual measured data nearby the port leads to the results:

- - Do not accurately capture perturbations due to nonlinear interactions due to processes such as refraction, diffraction and reflection
- - Underestimation of the results in the ocean surface wave.
- - Underestimation of current velocity.

In general, an adequate VMT simulation is achieved, at the spectral level it can be seen that the presence of the main waves (of higher spectral energy and those that possibly affected the ship) are consistent. The wavelets and Fourier transform show a correct trend of the wave characteristics measured in each sensor, as shown in Figure 9. Wavelets corresponding to both deep and shallow waters present several groups of waves forced by acoustic waves between 20-40 minutes, periods 40-60 min, and long-period waves about 120 min (Hu et al., 2023; Kubota et al., 2022; Omira et al., 2022). Likewise, the Fourier spectra also show two distinguished groups of 40-60 min and 100-120 min in shallow water.

Compared to the DART buoys, the simulations exhibit similar behaviors because they are located at great depths, where the influence of the bathymetry and the land boundaries is negligible. The simulations demonstrate that tide gauges reveal variations such as earlier arrival times or more energy, possibly associated with local effects and limitations in resolution. Despite these limitations, the model is sufficiently useful for the purpose of the study.

**COMMENT Nº 15:**

Lines 328 – 330: The extracted time series from the hydrodynamic model (and inputted to the mooring system) should be provided.

**RESPONSE TO COMMENT Nº15:**

*As you mentioned above, velocity is an important input. What can be provided is an analysis of it, indicating integral statistical parameters:*

The current velocity and water elevation time-series were extracted from the Boussinesq tsunami model (presented in Section 2.3 and validated in Section 3.2) at the location of Terminal 2 of La Pampilla port, coordinates -11.932 latitude, -77.181 longitude (Figure 11,a). The velocity of the current and Sea Water Elevation (SWE) input to the mooring line loading model are presented in Figure 11.c. The numerical results suggest that the ocean dynamics over time begins to be noticeable after 10 hours after the eruption, which coincides with the arrival of the leading Lamb wave and with maximums at times close to the time of the rupture accident according to the port authority. The maxima of the current and SWE date values close to 40 cm/s and 0.7 m respectively. On the other hand, the direction of the current suggests a north-south tilting motion, a similar pattern observed during the entire simulation.

The mooring scheme (Figure 11,b) is similar to the one found in Terminal 2 of the Port of La Pampilla in Peru during the mooring-break accident. . The vessel is moored to five buoys with eight moorings, one forward and four at the stern (in addition to two stern anchors anchored at a depth of 18 meters). The line number six (L6) was the one that broke in the actual accident that occurred at the Port of La Pampilla 15 hours after the eruption of the THTT volcano.

**COMMENT Nº 16:**

Line 329: The coordinates up to this point, including the maps, are given in decimal degrees, so they should be here with sufficient decimals (precision) considering the scale of the problem.

**RESPONSE TO COMMENT Nº16:**

*Thank you very much for your comment. We will change the coordinate format for this image. We hope that three decimal digits will be enough for the precision and scale of the problem:*

[Figure]

**Figure 11.** a, shows the location of the tide gauge **CL** and the port of La Pampilla at Callao Bay – Peru. b, is the berthing scheme implemented in the mooring lines loads model, where L6 is the line that broke and caused the instability in the mooring system safety. C$_{OG}$ is the center of gravity of the tanker and origin of the model coordinate system. c, is the time series extracted from the numerical model at tanker location and inputted in the mooring line load model: upper panel is the current velocity, the middle panel is the current direction and the lower panel is the variation of the sea water elevation (SWE).

**COMMENT Nº 17:**

Figure 12: A description of 'cspd' is needed in the caption.

**RESPONSE TO COMMENT Nº17:**

*Thank you very much for your comment. The definition is included in the description of the figure as follows:*

**Figure 12**. On the left is the initial layout with the origin at the ship's COG and on the right is the layout at the moment of mooring breakage, where the yellow and purple colors mark the maximum and minimum stress values, respectively. *cspd is the current speed at the moment of breaking of the mooring line*.

**COMMENT Nº 18:**

Lines 369-371: In my opinion, the sentence here should be modified because the reviewer understands the stress increase at specific mooring lines, and the authors intend to explain the potential reasons for this observation; however, if it is written like a direct conclusion rather than an observation and potential reasons, as is in the present form, then it is not clear how such conclusion is drawn and the analysis here seems not sufficient for this.

Original: "The increase in stresses was due to the configuration of the mooring layout, tsunami wave direction, and hydrodynamic effects, which could cause the mooring line to break."

Alternative The configuration of the mooring layout, tsunami wave direction, and hydrodynamic effects can be potential reasons for the increase in stresses, which could cause the mooring line to break.

**RESPONSE TO COMMENT Nº18:**

*Thanks for your suggestion, it is accurate. We have decided the first alternative:*

> The configuration of the mooring layout, tsunami wave direction, and hydrodynamic effects can be potential reasons for the increase in stresses, which could cause the mooring line to break. Secondly, the findings suggest that the VMT results in a significant increase in mooring stresses, exceeding 10-times the levels observed during the tsunami-only event (where the VMT is not included in the simulation).

**COMMENT Nº 19:**

Lines 373-375: "These results confirm that the atmospheric waves generated during the volcanic eruption have transmitted energy to the ocean, generating tsunami-like waves that have affected the far field." Various studies of the Tonga event from different perspectives have already confirmed such a finding and have this conclusion. It is recommended to be more specific to the findings of this study here, maybe focusing on the impact of VMT on marine vessels or other comments are encouraged.

**RESPONSE TO COMMENT Nº19:**

> *You are right. We can focus this result more on the impact of these waves on ships docked offshore in the far field. New sentence:*

> These results suggest that atmospheric waves generated during the volcanic eruption have triggered a VMT, generating tsunami-like waves that may have affected the mooring safety of vessels berthed in offshore ports in the far field.

**COMMENT Nº 20:**

The conclusion part could be stronger, highlighting the major findings in a more distinct manner and more discussions on the stress increase, and reasons for the amplification of amplitudes. **A thorough review and revision is recommended**.

**RESPONSE TO COMMENT Nº20:**

> The propagation of atmospheric waves and their coupling with the ocean were extensively studied following the Tonga event. Although epistemic uncertainties associated with the event are important, it was possible to understand the main drivers and effects of the volcano-meteorological tsunami (VMT) in the mooring safety of vessels moored in offshore ports in the far field. The air-ocean Proudman resonance in deep water was the driving mechanism that caused long period gravitational waves (which had very similar characteristics to those of a tectonic-source tsunami.) to travel along the Pacific Ocean and affect the offshore port La Pampilla in the far field.

> The potential of the explosion-induced atmospheric waves to magnify tsunamis once they pass over deep ocean regions has been evidenced. The second train of tsunami-like waves that arrived in the Peruvian coast and affected the mooring safety were generated due to Proudman resonance between acoustic-gravity waves with ocean-gravity waves, which produced an air–ocean energy transfer that led to an increase in tsunami energy in the far field instead of losing it. This happened near the Tonga trench in the propagation direction of South America (Figure 3-c in Omira et al., 2022).

> The presence of the high period waves exclusively in the tide gauges of Peru and Ecuador (CL and LA, respectively) could be explained by several processes associated with shoaling. These processes could include the width and slope of the continental shelf (which is wider on the Peruvian coast than on the Chilean coast, for example) and the effects associated with topographic boundaries and their geometry, such as the natural bay oscillation modes.

> The spectral analysis results establish the influence of the atmospheric waves generated by the HTHH volcano. Considering that, the 120-minutes long-period waves were associated with the air-ocean-resonance of the Tonga event and that numerical simulations additionally show the mooring line stress using the VMT time series, it is possible to conclude that the Tonga tsunami caused the overstressing and subsequent accident in the port of La Pampilla, Peru.

Hydrodynamic loads on the vessel's hull due to tsunamigenic phenomena can threaten the stability of moored vessels. These loads are mainly due to drag forces (driven by the tsunami kinetic energy) that affect ship stability. Based on the ship DOF, the results suggest that VMT affect mainly the horizontal plane motions, two associated with displacements (sway and surge) and the other with the angle of rotation about its vertical axis (yaw). This is likely explained by the large amount of kinetic energy that tsunamis have in their propagation, which travel at high speed, but with little wave elevation. The VMT produced during the Tonga 2022 event was accompanied by long-period waves and currents, which could affect the stability of the mooring system in the port of La Pampilla, Peru.

The hydrodynamic effects of very long period tsunami waves can generate damage similar to those of tsunamis of tectonic origin, affecting elements such as infrastructure, vessels, merchandise and people in port environments.

**Comments within the document**

**COMMENT № 21:**

Lines 1-2: either capital letters (for all words in the title) or small..please.

"Volcano Tsunamis  and their effects on moored vessels safety: The 2022 Tonga event".

**RESPONSE TO COMMENT №21:**

*Thank you very much for noticing this clear error. You will see corrected in the new document and highlighted in yellow in line 1 as:*

**"Volcano tsunamis and their effects on moored vessels safety: The 2022 Tonga event"**

**COMMENT № 22:**

Line 15: change in font style and improper capital letter detected.

rupture of the Vessel mooring lines occurred 15 hours after the eruption, resulting the spill of over 11,000

**RESPONSE TO COMMENT №22:**

*Again, thank you very much for noticing this clear error. You will see corrected in the new document.*

**COMMENT № 23:**

Lines 44-45: "Pacific. The latest eruptive phase of the HTHH volcano began in mid-December 2021 with vigorous shallow-water explosive activity".  Reference needed.

**RESPONSE TO COMMENT №23:**

The latest eruptive phase of the HTHH volcano began in mid-December 2021 with vigorous shallow-water explosive activity (Vergoz et al., 2022).

Vergoz, J., Hupe, P., Listowski, C., Le Pichon, A., Garcés, M. A., Marchetti, E., Labazuy, P., Ceranna, L., Pilger, C., Gaebler, P., Näsholm, S. P., Brissaud, Q., Poli, P., Shapiro, N., De Negri, R., and Mialle, P.: IMS observations of infrasound and acoustic-gravity waves produced by the January 2022 volcanic eruption of Hunga, Tonga: A global analysis, Earth Planet. Sci. Lett., 591, https://doi.org/10.1016/J.EPSL.2022.117639, 2022.

**COMMENT № 24:**

Lines 46-52: "Tonga experienced a volcano meteorological tsunami (VMT) following a violent volcanic explosion that generated atmospheric gravity waves that propagated several times across the globe (Omira et al., 2022; Wright et al., 2022). These waves resulted from particle agitation in the atmosphere, travelling both vertically and horizontally at sonic and supersonic speeds (Kubota et al., 2022; Matoza et

al., 2022; Wright et al., 2022; Dogan et al., 2023). Reports of ocean-free surface elevations greater than 1 m, causing damage to ports and infrastructure, emerged after the volcano explosion, originating from coastal areas near Tonga to the northwest and southeast Pacific".

This is not consistent with previous sentence if said as 'a violent volcanic explosion'. Please reconsider that sentence and the previous one for a fluent reading. Too long sentence; hard to read&understand. Please consider to split and rewrite.

**RESPONSE TO COMMENT Nº24:**

*New version:*

> The HTHH trigger a volcano meteorological tsunami (VMT) due to a volcanic explosion that generated atmospheric gravity waves that propagated several times across the globe (Omira et al., 2022; Wright et al., 2022). These atmospheric waves resulted from particle agitation in the atmosphere, travelling both vertically and horizontally at sonic and supersonic speeds (Kubota et al., 2022; Matoza et al., 2022; Wright et al., 2022; Dogan et al., 2023). Following the volcano explosion, there were reports of flooding more than 1 m, causing damage to ports and infrastructure in both near and far field (Ramírez-Herrera et al., 2022; Imamura et al., 2022)".

**COMMENT Nº 25:**

Lines 78-80: " Given the non-linearity of physical processes, it is pertinent to perform specific studies focused on each situation, configuration, and need (Zheng et al., 2022)." This sentence needs to be rewritten more specifically, including which processes and each situation, configuration and need of what? or regarding what?

**RESPONSE TO COMMENT Nº25:**

*New sentence*:

> Given the non-linearity of the hydromechanical and physical processes that involve stress studies in mooring systems, in which each ship has its own characteristics (geometric, inertial, among others), it is pertinent to perform specific studies focused on each situation, configuration, and need. (Zheng et al. 2022).

**COMMENT Nº 26:**

Lines 97-98**:** "Thirdly, we developed an analytical model to assess the stresses due to hydrodynamic loads on moored vessels". Could you be more specific within this sentence because as the reviewer understands, the scope of the proposed study is narrower than this.

**RESPONSE TO COMMENT Nº26:**

> Thirdly, we developed a model to assess the loads on vessel's mooring lines based on the rigid body analytical equations with six degrees of freedom (DOFs), this model uses as input the ocean dynamics (ocean elevation and velocities) caused by the VTM.

**COMMENT Nº 27:**

Lines 119-121: "The air pressure time-series shows a notable peak of approximately 2 hPa with the "N-wave" pulse shape associated with the leading Lamb wave (Lynett et al., 2022; Omira et al., 2022)". The reviewer suggests including more discussion on the observations of air pressure and sea waves, e.g., observed maximum amplitudes, their timing, etc.

**RESPONSE TO COMMENT Nº27:**

> The arrival time was less than 3 hours for the B1 sensor in the near field (shape differences are caused by the sample time), and between eight and twelve hours in the far field, about 10.000 km away. The air pressure time-series shows a notable peak of approximately 2 hPa with the "N-wave" pulse shape associated with the leading Lamb wave (Lynett et al., 2022; Omira et al.,

2022)". The seawater elevation, in deep waters Dart sensors, shows tsunami arrival times in two different moments related with the inverse barometer effect due to the leading lamb wave and the acoustic gravity waves coupled in ocean (VMT) respectively whit values no more than 10 cm. On the other hand, the tidal gauge shows tsunami waves values higher than 1 meter especially in the far field locations like Chilean and Peruvian coasts.